# SPD domain-specific batch normalization to crack interpretable unsupervised domain adaptation in EEG

**Reinmar J. Kobler**[1,2]**, Jun-ichiro Hirayama**[1]**, Qibin Zhao**[1]**, Motoaki Kawanabe**[1,2]

[1]RIKEN Center for Advanced Intelligence Project (RIKEN AIP), Tokyo, Japan
[2]Advanced Telecommunications Research Institute International (ATR), Kyoto, Japan
`kobler.reinmar@gmail.com, qibin.zhao@riken.jp`
`jun-ichiro.hirayama@a.riken.jp, kawanabe@atr.jp`

## Abstract

Electroencephalography (EEG) provides access to neuronal dynamics non-invasively with millisecond resolution, rendering it a viable method in neuroscience and healthcare. However, its utility is limited as current EEG technology does not generalize well across domains (i.e., sessions and subjects) without expensive supervised re-calibration. Contemporary methods cast this transfer learning (TL) problem as a multi-source/-target unsupervised domain adaptation (UDA) problem and address it with deep learning or shallow, Riemannian geometry aware alignment methods. Both directions have, so far, failed to consistently close the performance gap to state-of-the-art domain-specific methods based on tangent space mapping (TSM) on the symmetric, positive definite (SPD) manifold. Here, we propose a machine learning framework that enables, for the first time, learning domain-invariant TSM models in an end-to-end fashion. To achieve this, we propose a new building block for geometric deep learning, which we denote SPD domain-specific momentum batch normalization (SPDDSMBN). A SPDDSMBN layer can transform domain-specific SPD inputs into domain-invariant SPD outputs, and can be readily applied to multi-source/-target and online UDA scenarios. In extensive experiments with 6 diverse EEG brain-computer interface (BCI) datasets, we obtain state-of-the-art performance in inter-session and -subject TL with a simple, intrinsically interpretable network architecture, which we denote TSMNet. Code: https://github.com/rkobler/TSMNet.

## 1 Introduction

Electroencephalography (EEG) measures multi-channel electric brain activity from the human scalp with millisecond precision [1]. Transient modulations in the rhythmic brain activity can reveal cognitive processes [2], affective states [3] and a person's health status [4]. Unfortunately, these modulations exhibit low signal-to-noise ratio (SNR), domain shifts (i.e., changes in the data distribution) and have low specificity, rendering statistical learning a challenging task - particularly in the context of brain-computer interfaces (BCI) [5] where the goal is to predict a target from a short segment of multi-channel EEG data in real-time.

Under domain shifts, domain adaptation (DA), defined as learning a model from a source domain that performs well on a related target domain, offers principled statistical learning approaches with theoretical guarantees [6, 7]. DA in the BCI field mainly distinguishes inter-session and -subject transfer learning (TL) [8]. In inter-session TL, domain shifts, are expected across sessions mainly due to mental drifts (low specificity) as well as differences in the relative positioning of the electrodes and their impedances. Inter-subject TL is more difficult, as domain shifts are additionally driven by structural and functional differences in brain networks as well as variations in the performed task [9].

These domain shifts are traditionally circumvented by recording labeled calibration data and fitting domain-specific models [10, 11]. As recording calibration data is costly, models that are robust to

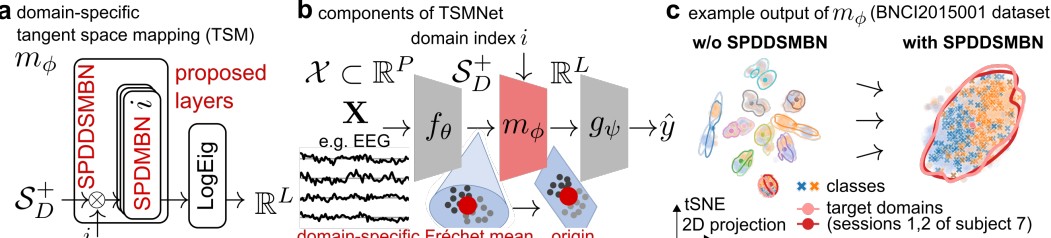

Figure 1: Visualization of the proposed framework around (**a**) SPD domain-specific momentum batch normalization (SPDDSMBN) that (**b**) learns parameters $\Theta = \{\theta, \phi, \psi\}$ of typical tangent space mapping (TSM) models *end-to-end* to crack multi-source/-target unsupervised domain adaptation on $\mathcal{S}_D^+$ for EEG data (**c**, illustrative example). For EEG data, we propose a simple, intrinsically interpretable parametrization of $f$ and $g$, denoted TSMNet, and obtain SoA performance.

scarce data with low SNR perform well in practice. Currently, tangent space mapping (TSM) models [12, 13] operating with symmetric, positive definite (SPD) covariance descriptors of preprocessed data are considered state-of-the-art (SoA) [10, 14, 15]. They are well suited for EEG data as they exhibit invariances to linear mixing of latent sources [16], and are consistent [13] and intrinsically interpretable [17] estimators for generative models that encode label information with a log-linear relationship in source power modulations.

Competitive, supervised calibration-free methods are one of the long-lasting grand challenges in EEG neurotechnology research [5, 10, 18, 19, 15]. Among the applied transfer learning techniques, including multi-task learning [20] and domain-invariant learning [21–23], unsupervised domain adaptation (UDA) [24] is considered as key to overcome this challenge [10, 19]. Contemporary methods cast the problem as a multi source and target UDA problem and address it with deep learning [25–28] or shallow, Riemannian geometry aware alignment methods [29–32]. Successful methods must cope with notoriously small and heterogeneous datasets (i.e., dozens of domains with a few dozens observations per domain and class). In a recent, relatively large scale inter-subject and -dataset TL competition with few labeled examples per target domain [19], deep learning approaches that aligned the first and second order statistics either in input [33, 27] or latent space [34] obtained the highest scores. Whereas, in a pure UDA competition [15] with a smaller dataset, Riemannian geometry aware approaches dominated. With the increasing popularity of geometric deep learning [35], Ju and Guan [36] proposed an architecture based on SPD neural networks [37] to align SPD features in latent space and attained SoA scores. Despite the tremendous advances in recent years, the field still lacks methods that can consistently close the performance gap to state-of-the-art domain-specific methods.

To close this gap, we propose a machine learning framework around domain-specific batch normalization on the SPD manifold (Figure 1). The proposed framework is used to implement domain-specific TSM (Figure 1a), which requires tracking the domains' Fréchet means in latent space as they are changing during training a typical TSM model in an end-to-end fashion (Figure 1b). After reviewing some preliminaries in section 2, we extend momentum batch normalization (MBN) [32] to SPDMBN that controls the Fréchet mean and variance of SPD data in section 3. In a theoretical analysis, we show under reasonable assumptions that SPDMBN can track and converge to the data's true Fréchet mean, enabling, for the first time, end-to-end learning of feature extractors, TSM and tangent space classifiers. Building upon this insight, we combine SPDMBN with domain-specific batch normalization (DSBN) [38] to form SPDDSMBN (Figure 1a). A SPDDSMBN layer can transform domain-specific SPD inputs into domain-invariant SPD outputs (Figure 1c). Like DSBN, SPDDSMBN easily extends to multi-source, multi-target and online UDA scenarios. In section 4, we briefly review the generative model of EEG, before the proposed methods are combined in a simple, intrinsically interpretable network architecture, denoted TSMNet (Figure 1b). We obtain state-of-the-art performance in inter-session and -subject UDA on small and large scale EEG BCI datasets, and show in an ablation study that the performance increase is primarily driven by performing DSBN on the SPD manifold.

## 2 Preliminaries

**Multi-source multi-target unsupervised domain adaptation**     Let $\mathcal{X}$ denote the space of input features, $\mathcal{Y}$ a label space, and $\mathcal{I}_d \subset \mathbb{N}$ an index set that contains unique domain identifiers. In

the multi-source, multi-target unsupervised domain adaptation scenario considered here, we are given a set $\mathcal{T}^{source} = \{\mathcal{T}_i | i \in \mathcal{I}_d^{source} \subset \mathcal{I}_d\}$ with $|\mathcal{I}_d^{source}| = N$ domains. Each domain $\mathcal{T}_i = \{(\mathbf{X}_{ij}, y_{ij})\}_{j=1}^{M} \sim P_{XY}^i$ contains $M$ observations of feature ($\mathbf{X} \in \mathcal{X}$) and label ($y \in \mathcal{Y}$) tuples sampled from a joint distribution $P_{XY}^i$.[1] While the joint distributions can be different (but related) across domains, we assume that the class priors are the same (i.e., $P_Y^i = P_Y$). The goal is to learn a predictive function $h : \mathcal{X} \times \mathcal{I}_d \to \mathcal{Y}$ that, once fitted to $\mathcal{T}^{source}$, can generalize to unseen target domains $\mathcal{T}^{target} = \{\mathcal{T}_l | l \in \mathcal{I}_d^{target} \subset \mathcal{I}_d, \mathcal{I}_d^{target} \cap \mathcal{I}_d^{source} = \emptyset\}$ merely based on *unsupervised* adaptation of $h$ to each target domain $\mathcal{T}_l$ once its label $l$ and features $\{\mathbf{X}_{lj}\}_{j=1}^{M} \sim P_X^l$ are revealed.

**Riemannian geometry on $\mathcal{S}_D^+$**   We start with recalling notions of geometry on the space of real $D \times D$ symmetric positive definite (SPD) matrices $\mathcal{S}_D^+ = \{\mathbf{Z} \in \mathbb{R}^{D \times D} : \mathbf{Z}^T = \mathbf{Z}, \mathbf{Z} \succ 0\}$. The space $\mathcal{S}_D^+$ forms a cone shaped Riemannian manifold in $\mathbb{R}^{D \times D}$ [39]. A Riemannian manifold $\mathcal{M}$ is a smooth manifold equipped with an inner product on the tangent space $\mathcal{T}_{\mathbf{Z}}\mathcal{M}$ at each point $\mathbf{Z} \in \mathcal{M}$. Tangent spaces have Euclidean structure with easy to compute distances $\mathcal{T}_{\mathbf{Z}}\mathcal{M} \times \mathcal{T}_{\mathbf{Z}}\mathcal{M} \to \mathbb{R}^+$ which locally approximate Riemannian distances on $\mathcal{M}$ induced by an inner product [40]. Logarithmic $\text{Log}_{\mathbf{Z}} : \mathcal{M} \to \mathcal{T}_{\mathbf{Z}}\mathcal{M}$ and exponential $\text{Exp}_{\mathbf{Z}} : \mathcal{T}_{\mathbf{Z}}\mathcal{M} \to \mathcal{M}$ mappings project points to and from tangent spaces.

Using the inner product $\langle \mathbf{S}_1, \mathbf{S}_2 \rangle_{\mathbf{Z}} = \text{Tr}(\mathbf{Z}^{-1}\mathbf{S}_1\mathbf{Z}^{-1}\mathbf{S}_2)$ for points $\mathbf{S}_1, \mathbf{S}_2$ in the tangent space $\mathcal{T}_{\mathbf{Z}}\mathcal{S}_D^+$ (i.e., the space of real symmetric $D \times D$ matrices) results in a globally defined affine invariant Riemannian metric on $\mathcal{S}_D^+$ [41, 39], which can be computed in closed form:

$$\delta_{AIRM}(\mathbf{Z}_1, \mathbf{Z}_2) = ||\log(\mathbf{Z}_1^{-\frac{1}{2}}\mathbf{Z}_2\mathbf{Z}_1^{-\frac{1}{2}})||_F \tag{1}$$

where $\mathbf{Z}_1$ and $\mathbf{Z}_2$ are two SPD matrices, $\log(\cdot)$ denotes the matrix logarithm[2], $||\cdot||_F$ the Frobenius norm, and $\text{Tr}(\cdot)$ in the inner product the trace operator. Due to affine invariance, we have $\delta(\mathbf{A}\mathbf{Z}_1\mathbf{A}^T, \mathbf{A}\mathbf{Z}_2\mathbf{A}^T) = \delta(\mathbf{Z}_1, \mathbf{Z}_2)$ for any invertible $D \times D$ transformation matrix $\mathbf{A}$. The exponential and logarithmic mapping are also globally defined in closed form as

$$\text{Log}_{\mathbf{Z}}(\mathbf{Z}_1) = \mathbf{Z}^{\frac{1}{2}}\log(\mathbf{Z}^{-\frac{1}{2}}\mathbf{Z}_1\mathbf{Z}^{-\frac{1}{2}})\mathbf{Z}^{\frac{1}{2}} \tag{2}$$

$$\text{Exp}_{\mathbf{Z}}(\mathbf{S}_1) = \mathbf{Z}^{\frac{1}{2}}\exp(\mathbf{Z}^{-\frac{1}{2}}\mathbf{S}_1\mathbf{Z}^{-\frac{1}{2}})\mathbf{Z}^{\frac{1}{2}} \tag{3}$$

For a set of SPD points $\mathcal{Z} = \{\mathbf{Z}_j \in \mathcal{S}_D^+\}_{j \leq M}$, we will use the notion of Fréchet mean $\mathbf{G}_{\mathcal{Z}} \in \mathcal{S}_D^+$ and Fréchet variance $\nu_{\mathcal{Z}}^2 \in \mathbb{R}^+$. The Fréchet mean is defined as the minimizer of the average squared distances

$$\mathbf{G}_{\mathcal{Z}} = \arg\min_{\mathbf{G} \in \mathcal{S}_D^+} \frac{1}{M} \sum_{j=1}^{M} \delta^2(\mathbf{G}, \mathbf{Z}_j) \tag{4}$$

For $M = 2$, there is a closed form solution expressed as

$$\mathbf{G}_{\mathcal{Z}}(\gamma) = \mathbf{Z}_1 \#_{\gamma} \mathbf{Z}_2 = \mathbf{Z}_1^{\frac{1}{2}} \left(\mathbf{Z}_1^{-\frac{1}{2}}\mathbf{Z}_2\mathbf{Z}_1^{-\frac{1}{2}}\right)^{\gamma} \mathbf{Z}_1^{\frac{1}{2}} \tag{5}$$

with weight $\gamma = 0.5$. Choosing $\gamma \in [0, 1]$ computes weighted means along the geodesic (i.e., the shortest curve) that connects both points. For $M > 2$, (4) can be solved using the Karcher flow algorithm [42], which iterates between projecting the data to the tangent space (2) at the current estimate, arithmetic averaging, and projecting the result back (3) to obtain a new estimate. The Fréchet variance $\nu_{\mathcal{Z}}^2$ is defined as the attained value at the minimizer $\mathbf{G}_{\mathcal{Z}}$:

$$\nu_{\mathcal{Z}}^2 = \text{Var}_{\mathcal{Z}}(\mathbf{G}_{\mathcal{Z}}) = \frac{1}{M} \sum_{j=1}^{M} \delta_{AIRM}^2(\mathbf{G}_{\mathcal{Z}}, \mathbf{Z}_j) \tag{6}$$

To shift a set of tangent space points to vary around a parametrized mean $\mathbf{G}_{\phi}$, parallel transport on $\mathcal{S}_D^+$ can be used [43]:

$$\Gamma_{\mathbf{G}_{\mathcal{Z}} \to \mathbf{G}_{\phi}}(\mathbf{S}) = \mathbf{E}^T\mathbf{S}\mathbf{E}, \quad \mathbf{E} = (\mathbf{G}_{\mathcal{Z}}^{-1}\mathbf{G}_{\phi})^{\frac{1}{2}} \tag{7}$$

While, parallel transport is generally defined for tangent space vectors $\mathbf{S}$ [40], on $\mathcal{S}_D^+$ the same operations also apply directly to points on the manifold (i.e., $\mathbf{Z} \in \mathcal{Z}$) [31, 44].

---

[1]For ease of notation, although not required by our method, we assume that $M$ is the same for each domain.
[2]For SPD matrices, powers, logarithms and exponentials can be computed via eigen decomposition.

Table 1: Overview and differences of relevant batch normalization algorithms. The last column, denoted normalization, sumarizes which statistics are used to normalize the batch data during training.

| Acronym | $\mathcal{S}_D^+$ | domain-specific | momentum $\gamma$ | normalization |
|---|---|---|---|---|
| MBN [32] | no | no | adaptive | running stats |
| SPDBN [46] | yes | no | fixed | running stats |
| SPDMBN (algorithm 1) *proposed* | yes | no | adaptive | running stats |
| DSBN [38] | no | yes | fixed | batch stats |
| SPDDSMBN (18) *proposed* | yes | yes | adaptive | running stats |

# 3  Domain-specific batch normalization on $\mathcal{S}_D^+$

In this section, we review relevant batch normalization (BN) [45] variants with a focus on $\mathcal{S}_D^+$. We then present SPDMBN and show in a theoretical analysis that the running estimate converges to the true Fréchet mean under reasonable assumptions. At last, we combine the idea of domain-specific batch normalization (DSBN) [38] with SPDMBN to form a SPDDSMBN layer. Table 1 provides a brief overview of related and proposed methods.

**Batch normalization**    Batch normalization (BN) [45] is a widely used training technqiue in deep learning as BN layers speed up convergence and improve generalization via smoothing of the engery landscape [47, 32]. A standard BN layer applies slightly different transformations during training and testing to independent and identically distributed (iid) observations $\mathbf{x}_j \in \mathbb{R}^d$ within the $k$-th minibatch $\mathcal{B}_k$ of size $M$ drawn from a dataset $\mathcal{T}$. During training, the data are normalized using the batch mean $\mathbf{b}_k$ and variance $\mathbf{s}_k^2$, and then scaled and shifted to have a parametrized mean $\mathbf{g}_\phi$ and variance $\boldsymbol{\sigma}_\phi^2$. Internally, the layer updates running estimates of the dataset's statistics $(\mathbf{g}_k, \boldsymbol{\sigma}_k^2)$ during each training step $k$; the updates are computed via exponential smoothing with momentum parameter $\gamma$. During testing, the running estimates are used.

Using batch statistics to normalize data during training rather than running estimates introduces noise whose level depends on the batch size [32]; smaller batch sizes raise the noise level. The introduced noise regularizes the training process, which can help to escape poor local minima in initial learning but also lead to underfitting. Momentum BN (MBN) [32] allows small batch sizes while avoiding underfitting. Like batch renormalization [48], MBN uses running estimates during training and testing. The key difference is that MBN keeps two sets of running statistics; one for training and one for testing. The latter are updated conventionally, while the former are updated with momentum parameter $\gamma_{train}(k)$ that decays over training steps $k$. MBN can, therefore, quickly escape poor local minima during initial learning and avoid underfitting at later stages [32].

**Batch normalization on $\mathcal{S}_D^+$**    It is intractable to compute the Fréchet mean $\mathbf{G}_{\mathcal{B}_k}$ for each minibatch $\mathcal{B}_k = \{\mathbf{Z}_j \in \mathcal{S}_D^+\}_{j=1}^M$, as there is no efficient algorithm to solve (4). Brooks et al. [44] proposed Riemannian Batch Normalization (RBN) as a tractable approximation. RBN approximateley solves (4) by aborting the iterative Karcher flow algorithm after one iteration. To transform $\mathbf{Z}_j \in \mathcal{B}_k$ with estimated mean $\mathbf{B}_k$ to vary around $\mathbf{G}_\phi$, parallel transport (7) is used. The RBN input output transformation is then expressed as

$$\text{RBN}(\mathbf{Z}_j; \mathbf{G}_\phi, \gamma) = \Gamma_{\mathbf{B}_k \to \mathbf{G}_\phi}(\mathbf{Z}_j) = \mathbf{E}^T \mathbf{Z}_j \mathbf{E} , \quad \mathbf{E} = (\mathbf{B}_k^{-1} \mathbf{G}_\phi)^{\frac{1}{2}} , \quad \forall \mathbf{Z}_j \in \mathcal{B}_k \qquad (8)$$

Using (5), the running estimate of the dataset's Fréchet mean can be updated in closed form

$$\mathbf{G}_k = \mathbf{G}_{k-1} \#_\gamma \mathbf{B}_k \qquad (9)$$

In [46] we proposed an extension to RBN, denoted SPD batch renormalization (SPDBN) that controls both Fréchet mean and variance. Like batch renormalization [48], SPDBN uses running estimates $\mathbf{G}_k$ and $\nu_k^2$ during training and testing. To transform $\mathbf{Z}_j \in \mathcal{B}_k$ to vary around $\mathbf{G}_\phi$ with variance $\nu_\phi^2$, each observation is first transported to vary around the identity matrix $\mathbf{I}$, rescaled via computing matrix powers and finally transported to vary around $\mathbf{G}_\phi$. The sequence of operations can be expressed as

$$\text{SPDBN}(\mathbf{Z}_j; \mathbf{G}_\phi, \nu_\phi^2, \varepsilon, \gamma) = \Gamma_{\mathbf{I} \to \mathbf{G}_\phi} \circ \Gamma_{\mathbf{G}_k \to \mathbf{I}}(\mathbf{Z}_j)^{\frac{\nu_\phi}{\nu_k + \varepsilon}} , \quad \forall \mathbf{Z}_j \in \mathcal{B}_k \qquad (10)$$

The standard backpropagation framework with extensions for structured matrices [49] and manifold-constrained gradients [40] can be used to propagate gradients through RBN and SPDBN layers and learn the parameters $(\mathbf{G}_\phi, \nu_\phi)$.

**Algorithm 1:** SPD momentum batch normalization (SPDMBN)

---

**Input** : batch $\mathcal{B}_k = \{\mathbf{Z}_j \in \mathcal{S}_D^+\}_{j=1}^M$ at training step $k$,
running mean $\bar{\mathbf{G}}_{k-1}$, $\bar{\mathbf{G}}_0 = \mathbf{I}$ and variance $\bar{\nu}_{k-1}^2$, $\bar{\nu}_0^2 = 1$ for training,
running mean $\tilde{\mathbf{G}}_{k-1}$, $\tilde{\mathbf{G}}_0 = \mathbf{I}$ and variance $\tilde{\nu}_{k-1}^2$, $\tilde{\nu}_0^2 = 1$ for testing,
learnable parameters $(\mathbf{G}_\phi, \nu_\phi)$, and momentum for training and testing $\gamma_{train}(k), \gamma \in [0,1]$

**Output** : normalized batch $\{\tilde{\mathbf{Z}}_j = \mathrm{SPDMBN}(\mathbf{Z}_j) \in \mathcal{S}_D^+ \mid \mathbf{Z}_j \in \mathcal{B}_k\}$

**if** `training` **then**
    $\mathbf{B}_k = \texttt{karcher\_flow}\,(\mathcal{B}_k, \texttt{steps} = 1);$              // approx. batch Fréchet mean
    $\bar{\mathbf{G}}_k = \bar{\mathbf{G}}_{k-1} \#_{\gamma_{train}(k)} \mathbf{B}_k;$                  // update running stats
    $\bar{\nu}_k^2 = (1 - \gamma_{train}(k))\bar{\nu}_{k-1}^2 + \gamma_{train}(k)\mathrm{Var}_{\mathcal{B}_k}(\bar{\mathbf{G}}_k)$
    $\tilde{\mathbf{G}}_k = \tilde{\mathbf{G}}_{k-1} \#_\gamma \mathbf{B}_k$
    $\tilde{\nu}_k^2 = (1 - \gamma)\tilde{\nu}_{k-1}^2 + \gamma\mathrm{Var}_{\mathcal{B}_k}(\tilde{\mathbf{G}}_k)$
**end**
$(\mathbf{G}_k, \nu_k^2) = (\bar{\mathbf{G}}_k, \bar{\nu}_k)$ **if** `training` **else** $(\tilde{\mathbf{G}}_k, \tilde{\nu}_k^2)$
$\tilde{\mathbf{Z}}_j = \Gamma_{\mathbf{I} \to \mathbf{G}_\phi} \circ \Gamma_{\mathbf{G}_k \to \mathbf{I}}(\mathbf{Z}_j)^{\frac{\nu_\phi}{\nu_k + \varepsilon}};$    // parallel transport to whiten, rescale and rebias

---

**Momentum batch normalization on $\mathcal{S}_D^+$**    SPDBN [46] suffers from the same limitations as batch renormalization [48]. Consequently, we propose to extend MBN [32] to $\mathcal{S}_D^+$. We list the pseudocode of our proposed extension, which we denote SPDMBN, in algorithm 1. SPDMBN uses approximations of batch-specific Fréchet means to update two sets of running estimates of the dataset's Fréchet mean. As MBN [32], we decay $\gamma_{train}(k)$ with a clamped exponential decay schedule

$$\gamma_{train}(k) = 1 - \gamma_{min}^{\frac{1}{K-1}\max(K-k,0)} + \gamma_{min} \tag{11}$$

where $K$ defines the training step at which $\gamma_{min} \in [0,1]$ should be attained.

**The running mean in SPDMBN converges to the Fréchet mean**    Here, we consider models that apply a SPDMBN layer to latent representations generated by a feature extractor $f_\theta : \mathcal{X} \to \mathcal{S}_D^+$ with learnable parameters $\theta$.

We define a dataset that contains the latent representations generated with feature set $\theta_k$ as $\mathcal{Z}_{\theta_k} = \{f_{\theta_k}(\mathbf{x}) | \mathbf{x} \in \mathcal{T}\}$, and a minibatch of $M$ iid samples at training step $k$ as $\mathcal{B}_k$. We denote the Fréchet mean of $\mathcal{Z}_{\theta_k}$ as $\mathbf{G}_{\theta_k}$, the estimated Fréchet mean, defined in (9), as $\mathbf{G}_k$, and the estimated batch mean as $\mathbf{B}_k$. Since the batches are drawn randomly, we consider the batch and running means as random variables.

We assume that the variance $\mathrm{Var}_{\theta_k}(\mathbf{B}_{k-1}) = \mathbb{E}_{\mathbf{B}_{k-1}}\{\delta^2(\mathbf{B}_{k-1}, \mathbf{G}_{\theta_k})\}$ of the previous batch mean $\mathbf{B}_{k-1}$ with respect to the current Fréchet mean $\mathbf{G}_{\theta_k}$ is bounded by the current variance $\mathrm{Var}_{\theta_k}(\mathbf{B}_k)$ and the norm of the difference in the parameters

$$\mathrm{Var}_{\theta_k}(\mathbf{B}_{k-1}) \leq (1 + ||\theta_k - \theta_{k-1}||)\mathrm{Var}_{\theta_k}(\mathbf{B}_k) \tag{12}$$

That is, across training steps $k$ the parameter updates are required to change the first and second order moments of the distribution of $\mathcal{Z}_{\theta_k}$ gradually so that the expected distance between $\mathbf{G}_{\theta_k}$ and the previous batch mean $\mathbf{B}_{k-1}$ is bounded. We conjecture that this is the case for feature extractors $f_\theta$ that are smooth in the parameters and small learning rates, but leave the proof for future work.

**Proposition 1** (Error bound for $\mathbf{G}_k$). *Consider the setting defined above, and assumption (12) holds true. Then, the variance of the running mean $\mathrm{Var}_{\theta_k}(\mathbf{G}_k)$ is bounded by*

$$\mathrm{Var}_{\theta_k}(\mathbf{G}_k) \leq \mathrm{Var}_{\theta_k}(\mathbf{B}_k) \tag{13}$$

*over training steps $k$ if*

$$||\theta_k - \theta_{k-1}|| \leq \frac{1 - \gamma^2}{(1-\gamma)^2} - 1 \tag{14}$$

*holds true.*

The proof is provided in appendix A.1 of the supplementary material and relies on the proof of the geometric law of large numbers [50].

Proposition 1 states that if (12) and (14) are met, the expected distance between the true Fréchet mean and the running mean is less or equal to the one of the batch mean. Consequently, the introduced

noise level of SPDBN (equation 10) and SPDMBN (algorithm 1), which use $\mathbf{G}_k$ to normalize batches during training, is smaller or equal to RBN (equation 8), which uses $\mathbf{B}_k$.

Since $\gamma$ controls the adaptation speed of $\mathbf{G}_k$, proposition 1 also states that if $\gamma$ converges to zero (=no adaptation), the parameter updates are required to converge to zero as well (=no learning). Hence, for a fixed $\gamma \in (0, 1)$, as in the case of SPDBN (equation 10), proposition 1 is fulfilled, if the learning rate for the parameters $\theta$ is chosen sufficiently small. This can substantially slow down initial learning for standard choices of $\gamma$ (e.g., 0.1 or 0.01). As a remedy, SPDMBN (algorithm 1) uses an adaptive momentum parameter, which allows larger parameter updates during initial training steps.

If we consider a late stage of learning, and in particular assume that after a certain number of iterations $\kappa$ the parameters stay in a small ball with radius $\rho$ around $\theta^*$ (i.e., $||\theta_k - \theta^*|| \leq \rho \ \ \forall \, k > \kappa$) and the feature extractor is $L$-smooth in the parameters (i.e., $\delta(f_\theta(\mathbf{x}), f_{\tilde{\theta}}(\mathbf{x})) \leq L||\theta - \tilde{\theta}|| \ \forall \mathbf{x} \in \mathcal{T}$, $\forall \theta, \tilde{\theta}$) then the distances are bounded $\delta(f_{\theta_k}(\mathbf{x}), f_{\theta^*}(\mathbf{x})) \leq \rho L$.

**Remark 1** (Convergence of $\mathbf{G}_k$ for SPDMBN). If $\rho L$ is neglibile compared to the dataset's variance, then the Fréchet mean and variance can be considered fixed, and the theorem of large numbers on $\mathcal{S}_D^+$ [50] applies directly. That is, if the momentum parameter is decayed exponentially $\forall k > \kappa$ the running mean $\mathbf{G}_k$ converges to the Fréchet mean $\mathbf{G}_{\theta^*}$ in probability as $k \to \infty$.

Taken together, Proposition 1 and Remark 1 provide guidelines to update $\mathbf{G}_k$ in SPDMBN so that the introduced estimation error is bounded during initial fast learning (large $\gamma$) and decays towards zero in late learning (small $\gamma$).

**SPDMBN to learn tangent space mapping at Fréchet means** Typical TSM models for classification [12] and regression [13] first use (2) to project $\mathbf{Z} \in \mathcal{T} \subset \mathcal{S}_D^+$ to the tangent space at the Fréchet mean $\mathbf{G}_{\mathcal{T}}$, then use (7) to transport the result to vary around $\mathbf{I}$, and finally extract elements in the upper triangular part[3] to reduce feature redundancy. The invertible mapping $\mathcal{P}_{\mathbf{G}_{\mathcal{T}}} : \mathcal{S}_D^+ \to \mathbb{R}^{D(D+1)/2}$ is expressed as:

$$\mathcal{P}_{\mathbf{G}_{\mathcal{T}}}(\mathbf{Z}) = \text{upper} \circ \Gamma_{\mathbf{G}_{\mathcal{T}} \to \mathbf{I}} \circ \text{Log}_{\mathbf{G}_{\mathcal{T}}}(\mathbf{Z}) = \text{upper}(\log(\mathbf{G}_{\mathcal{T}}^{-\frac{1}{2}} \mathbf{Z} \mathbf{G}_{\mathcal{T}}^{-\frac{1}{2}})) \tag{15}$$

We propose to use a SPDMBN layer followed by a LogEig layer [37] to compute a similar mapping $m_\phi$ (Figure 1a). A LogEig layer simply computes the matrix logarithm and vectorizes the result so that the norm is preserved. If the parametrized mean of SPDMBN is fixed to the identify matrix ($\mathbf{G}_\phi = \mathbf{I}$), the composition computes

$$m_\phi(\mathbf{Z}) = \text{LogEig} \circ \text{SPDMBN}(\mathbf{Z}) = \text{upper} \circ \log \circ \Gamma_{\mathbf{G}_k \to \mathbf{I}}(\mathbf{Z})^{\frac{\nu_\phi}{\nu_k + \varepsilon}}$$
$$= \text{upper}\left(\frac{\nu_\phi}{\nu_k + \varepsilon} \log\left(\mathbf{G}_k^{-\frac{1}{2}} \mathbf{Z} \mathbf{G}_k^{-\frac{1}{2}}\right)\right) \tag{16}$$

where $(\mathbf{G}_k, \nu_k^2)$ are the estimated Fréchet mean and variance of the dataset $\mathcal{T}$ at training step $k$, and $\phi = \{\nu_\phi\}$ the learnable parameters. According to remark 1 $\mathbf{G}_k$ converges to $\mathbf{G}_{\mathcal{T}}$ and, in turn, $m_\phi$ to a scaled version of $\mathcal{P}_{\mathbf{G}_{\mathcal{T}}}$, since upper is linear.

The mapping $m_\phi$ offers several advantageous properties. First, the features are projected to a Euclidean vector space where standard layers can be applied and distances are cheap to compute. Second, distances between the projected features locally approximate $\delta$ and, therefore, inherit its invariance properties (e.g., affine mixing) [41]. This improves upon a LogEig layer [37] which projects features to the tangent space at the identity matrix. As a result, distances between LogEig projected features correspond to distances measured with the log-Euclidean Riemannian metric (LERM) [51] which is not invariant to affine mixing. Third, controlling the Fréchet variance in (16) empirically speeds up learning and improves generalization [46].

**Domain-specific batch normalization on $\mathcal{S}_D^+$** Considering a multi-source UDA scenario, Chang et al. [38] proposed a domain-specific BN (DSBN) layer which simply keeps multiple parallel BN layers and distributes observations according to the associated domains. Formally, we consider minibatches $\mathcal{B}_k$ that form the union of $N_{\mathcal{B}_k} \leq |\mathcal{I}_d|$ domain-specific minibatches $\mathcal{B}_k^i$ drawn from distinct domains $i \in \mathcal{I}_{\mathcal{B}_k} \subseteq \mathcal{I}_d$. As before, each $\mathcal{B}_k^i$ contains $j = 1, ..., M/N_{\mathcal{B}_k}$ iid observations $\mathbf{x}_j$. A DSBN layer mapping $\mathbb{R}^d \times \mathcal{I}_d \to \mathbb{R}^d$ can then be expressed as

$$\text{DSBN}(\mathbf{x}_j, i) = \text{BN}_i(\mathbf{x}_j; \mathbf{g}_{\phi_i}, \mathbf{s}_{\phi_i}, \varepsilon, \gamma) , \quad \forall \mathbf{x}_j \in \mathcal{B}_k^i , \quad \forall i \in \mathcal{I}_{\mathcal{B}_k} \tag{17}$$

---

[3]To preserve the norm, the off diagonal elements are scaled by $\sqrt{2}$.

In practice, the batch size $M$ is typically fixed. The particular choice is influenced by resource availability and the desired noise level introduced by minibatch based stochastic gradient descent. A drawback of DSBN is that for a fixed batch size $M$ and an increasing number of source domains $N_{\mathcal{B}_k}$, the effective batch size declines for the BN layers within DSBN. Since small batch sizes increase the noise level introduced by BN, increasing the number of domains per batch can lead to underfitting [32]. To alleviate this effect, we use the previously introduced SPDMBN layer. The proposed domain-specific BN layer on $\mathcal{S}_D^+$ is then formally defined as

$$\text{SPDDSMBN}(\mathbf{Z}_j, i) = \text{SPDMBN}_i(\mathbf{Z}_j; \mathbf{G}_{\phi_i}, \nu_{\phi_i}, \varepsilon, \gamma, \gamma_{train}(k)) \,, \; \forall \mathbf{Z}_j \in \mathcal{B}_k^i \subset \mathcal{S}_D^+ \,, \; \forall i \in \mathcal{I}_{\mathcal{B}_k} \tag{18}$$

The layer can be readily adapted to new domains, as new SPDMBN layers can be added on the fly. If the entire data of a domain becomes available, the domain-specific Fréchet mean and variance can be estimated by solving (4), otherwise, the update rules in algorithm 1 can be used.

## 4 SPDDSMBN to crack interpretable multi-source/-target UDA for EEG data

With SPDDSMBN introduced in the previous section, we focus on a specific application domain, namely, multi-source/-target UDA for EEG-based BCIs and propose an intrinsically interpretable architecture which we denote TSMNet.

**Generative model of EEG** EEG signals $\mathbf{x}(t) \in \mathbb{R}^P$ capture voltage fluctuations on $P$ channels. An EEG record (=domain) is uniquely identified by a subject and session identifier. After standard pre-processing steps, each domain $i$ contains $j = 1, ..., M$ labeled observations with features $\mathbf{X}_{ij} \in \mathcal{X} \subset \mathbb{R}^{P \times T}$ where $T$ is the number of temporal samples. Due to linearity of Maxwell's equations and Ohmic conductivity of tissue layers in the frequency ranges relevant for EEG [52], a domain-specific linear instantaneous mixture of sources model is a valid generative model:

$$\mathbf{X}_{ij} = \mathbf{A}_i \mathbf{S}_{ij} + \mathbf{N}_{ij} \tag{19}$$

where $\mathbf{S}_{ij} \in \mathbb{R}^{Q \times T}$ represents the activity of $Q$ latent sources, $\mathbf{A}_i \in \mathbb{R}^{P \times Q}$ a domain-specific mixing matrix and $\mathbf{N}_{ij} \in \mathbb{R}^{P \times T}$ additive noise. Both $\mathbf{A}_i$ and $\mathbf{S}_{ij}$ are unknown which demands making assumptions on $\mathbf{A}_i$ (e.g., anatomical prior knowledge [53]) and/or $\mathbf{S}_{ij}$ (e.g., statistical independence [54]) to extract interesting sources.

**Interpretable multi-source/-target UDA for EEG data** As label information is available for the source domains, our goal is to identify discriminative oscillatory sources shared across domains. Our approach relies on TSM models with linear classifiers [12], as they are consistent [13] and intrinsically interpretable [17] estimators for generative models with log-linear relationships between the target $y_{ij}$ and variance $\text{Var}\{s_{ij}^{(k)}(t)\}$ of $k = 1, ..., K \leq Q$ discriminative sources:

$$y_{ij} = \sum_{k=1}^{K} b_k \log\left(\text{Var}\{s_{ij}^{(k)}(t)\}\right) + \varepsilon_{ij} \tag{20}$$

where $b_k \in \mathbb{R}$ summarizes the coupling between the target $y_{ij}$ and the variance of the encoding source, and $\varepsilon_{ij}$ additive noise. In [17] we showed that the encoding sources' coupling and their patterns[4] (columns of $\mathbf{A}_i$) can be recovered via solving a generalized eigenvalue problem between the Fréchet mean $\mathbf{G}_{\mathcal{T}_i}$ and classifier patterns [55] that were back projected to $\mathcal{S}_D^+$ with $\mathcal{P}_{\mathbf{G}_{\mathcal{T}_i}}^{-1}$. The resulting eigenvectors are the patterns and the eigenvalues $\lambda_k$ reflect the relative source contribution $c_k$:

$$c_k = \max(\lambda_k, \lambda_k^{-1}) \,, \quad \lambda_k = \exp(b_k / ||\mathbf{b}||_2^2) \tag{21}$$

To benefit from the intrinsic interpretability of TSM models, we constrain our hypothesis class $\mathcal{H}$ to functions $h : \mathcal{X} \times \mathcal{I}_d \to \mathcal{Y}$ that can be decomposed into a composition of a shared linear feature extractor with covariance pooling $f_\theta : \mathcal{X} \to \mathcal{S}_D^+$, domain-specific tangent space mapping $m_\phi : \mathcal{S}_D^+ \times \mathcal{I}_d \to \mathbb{R}^{D(D+1)/2}$, and a shared linear classifier $g_\psi : \mathbb{R}^{D(D+1)/2} \to \mathcal{Y}$ with parameters $\Theta = \{\theta, \phi, \psi\}$.

---

[4]Here, we use the entire dataset's Fréchet mean instead of the domain-specific ones to compute patterns for the average domain.

**TSMNet with SPDDSMBN** Unlike previous approaches which learn $f_\theta, m_\phi, g_\psi$ sequentially [29, 31, 13, 30], we parametrize $h = g_\psi \circ m_\phi \circ f_\theta$ as a neural network and learn the entire model in an end-to-end fashion (Figure 1b). Details of the proposed architecture, denoted TSMNet, are provided in appendix A.2.5. In a nutshell, we parametrize $f_\theta$ as the composition of the first two linear convolutional layers of ShConvNet [56], covariance pooling [57], BiMap [37], and ReEig [37] layers. A BiMap layer applies a linear subspace projection, and a ReEig layer thresholds eigenvalues of symmetric matrices so that the output is SPD. We used the default threshold ($10^{-4}$) and found that it was never active in the trained models. Hence, after training, $f_\theta$ fulfilled the hypothesis class constraints. In order for $m_\phi$ to align the domain data and compute TSM, we use SPDDSMBN (18) with shared parameters (i.e., $\mathbf{G}_{\phi_i} = \mathbf{G}_\phi = \mathbf{I}, \nu_{\phi_i} = \nu_\phi$) in (16). Finally, the classifier $g_\psi$ was parametrized as a linear layer with softmax activations. We use the standard-cross entropy loss as training objective, and optimized the parameters with the Riemannian ADAM optimizer [58].

## 5 Experiments with EEG data

In the following, we apply our method to classify target labels from short segments of EEG data. We consider two BCI applications, namely, mental imagery [59, 5] and mental workload estimation [60]. Both applications have high potential to aid society in rehabilitation and healthcare [61, 62, 18] but have, currently, limited practical value because of poor generalization across sessions and subjects [19, 15].

**Datasets and preprocessing** The considered mental imagery datasets were BNCI2014001 [63] (9 subjects/2 sessions/4 classes), BNCI2015001 [64] (12/2-3/2), Lee2019 [65] (54/2/2), Lehner2020 [66] (1/7/2), Stieger2021 [67] (62/4-8/4) and Hehnberger2021 [68] (1/26/4). For mental workload estimation, we used a recent competition dataset [69] (12/2/3). A detailed description of the datasets is provided in appendix A.2.1. Altogether, we analyzed a total of 603 sessions of 158 human subjects whose data was acquired in previous studies that obtained the subjects' informed consent and the right to share anonymized data.
The python packages moabb [14] and mne [70] were used to preprocess the datasets. The applied steps comprise resampling the EEG signals to 250/256 Hz, applying temporal filters to extract oscillatory EEG activity in the 4 to 36 Hz range (spectrally resolved if required by a method) and finally extract short segments ($\leq 3s$) associated to a class label (details provided in appendix A.2.2).

**Evaluation** We evaluated TSMNet against several baseline methods implementing direct transfer or multi-source (-target) UDA strategies. They can be broadly categorized as component based [71, 68], Riemannian geometry aware [12, 17, 30, 72] or deep learning [56, 73, 25]. All models were fit and evaluated with a randomized leave 5% of the sessions (inter-session TL) or subjects (inter-subject TL) out cross-validation (CV) scheme. For inter-session TL, the models were only provided with data of the associated subject. When required, inner train/test splits (neural nets) or CV (shallow methods) were used to optimize hyper parameters (e.g., early stopping, regularization parameters). The dataset Hehenberger2021 was used to fit the hyper parameters of TSMNet, and is, therefore, omitted in the presented results. Balanced accuracy (i.e., the average recall across classes) was used as scoring metric. As the discriminability of the data varies considerably across subjects, we decided to report the results in the figures relative to the score of a SoA domain-specific Riemannian geometry aware method [17], which was fitted and evaluated in a 80%/20% chronological train/test split (for details see appendix A.2.4).

**Soft- and hardware** We either used publicly available python code or implemented the methods in python using the packages torch [74], scikit-learn [75], skorch [76], geoopt [77], mne [70], pyriemann [78], pymanopt [79]. We ran the experiments on standard computation PCs equipped with 32 core CPUs with 128 GB of RAM and used up to 1 GPU (24 GRAM). Depending on the dataset size, fitting and evaluating TSMNet varied from a few seconds to minutes.

### 5.1 Mental imagery
**TSMNet closes the gap to domain-specific methods** Figure 2 summarizes the mental imagery results. It displays test set scores of the considered TL methods relative to the score of a SoA domain-specific reference method. Combining the results of all subjects across datasets (Figure 2a), it becomes apparent that TSMNet is the only method that can significantly reduce the gap to the reference method (inter-subject) or even exceed its performance (inter-session). Figure 2b displays the results resolved across datasets (for details see appendix A.3.1). We make two important observations. First, concerning inter-session TL, TSMNet meets or exceeds the score of the reference method

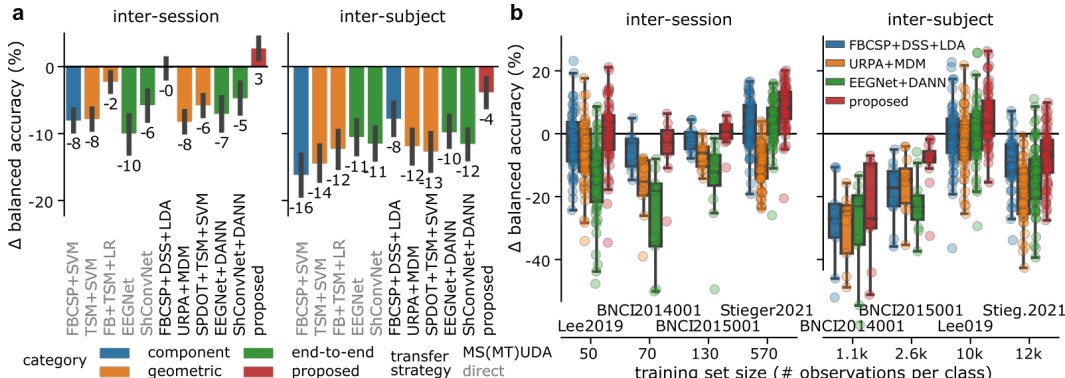

Figure 2: Mental imagery results (5 datasets). BCI test set score (balanced accuracy) for inter-session/-subject transfer learning methods relative to a SoA domain-specific reference model (80%/20% chronological train/test split; for details see appendix A.2.4). **a**, Barplots summarize the grand average (573 sessions, 138 subjects) results. Errorbars indicate bootstrapped (1e3 repetitions) 95% confidence intervals (over subjects). **b**, Box and scatter plots summarize the dataset-specific results for selected methods from each category. Datasets are ordered according to the training set size. Each dot summarizes the score for one subject. Lehner2021 is not displayed as it contains only 1 subject.

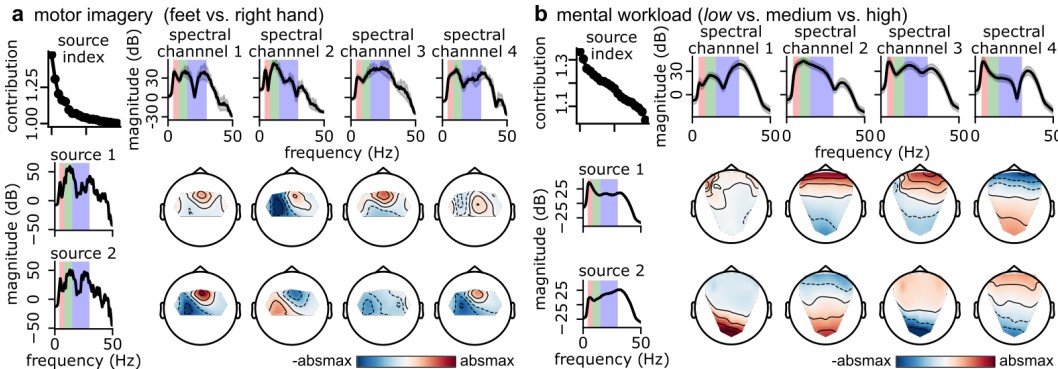

Figure 3: Model interpretation results. Patterns extracted from a TSMNet. **a**, Motor imagery dataset (BNCI2015001, inter-subject TL). The top, left panel lists the contribution, defined in (21), for each extracted source $k = 1, ..., 20$ (x-axis) to the target class. Panels in the left column summarize the spectral patterns of extracted sources. For visualization purposes, only the 2 most discriminative sources are displayed. Panels in the top row summarize the frequency profile of each spectral channel (output of 4 temporal convolution layers in $f_\theta$). Topographic plots summarize how the source activity is projected to regions covered by the EEG channels (rows correspond to the source index; columns to spectral channels). EEG channels at darker blue or red areas capture more source activity and are, therefore, more discriminative. **b**, As in **a** for the mental workload estimation dataset and class *low*.

consistently across datasets. Second, concerning inter-subject TL, we found that all considered methods tend to reduce the performance gap as the dataset size (# subjects) increases, and that TSMNet is consistently the top or among the top methods. As a fitted TSMNet corresponds to a typical TSM model with a linear classifier, we can transform the fitted parameters into interpretable patterns [17]. Figure 3a displays extracted patterns for the BNCI2015001 dataset (inter-subject TL). It is clearly visible that TSMNet infers the target label from neurophysiologically plausible sources (rows in Figure 3a). As expected [2], the source with highest contribution has spectral peaks in the alpha and beta bands, and originates in contralateral and central sensorimotor cortex.

**DSBN on $\mathcal{S}_D^+$ drives the success of TSMNet** Since TSMNet combines several advances, we present the results of an ablation study in Table 2. It summarizes the grand average inter-session TL test scores relative to TSMNet with SPDDSMBN for n = 138 subjects. We observed three significant effects. The largest effect can be attributed to $\mathcal{S}_D^+$, as we observed the largest performance decline if the architecture would be modified[5] to omit the SPD manifold (4.5% with DSBN, 3% w/o DSBN).

---

[5]We replaced the covariance pooling, BiMap, ReEig, SPD(DS)MBN, LogEig layers with variance pooling, elementwise log activations followed by (DS)MBN. Note that the resulting architecture is similar to ShConvNet.

Table 2: Ablation study. Grand average (5 mental imagery datasets, 138 subjects, inter-session TL) score for the test data relative to the proposed method, and training fit time (50 epochs). Standard-deviation is used to report the variability across subjects. Permutation t-tests (1e4 perms, df=137, 4 tests with t-max adjustment) were used to identify significant effects.

| | | | $\Delta$ balanced accuracy (%) | | fit time (s) |
|---|---|---|---|---|---|
| $\mathcal{S}_D^+$ | DSBN [1] | BN method | mean (std) | t-val (p-val) | mean (std) |
| yes | yes | SPDMBN (*proposed*) | - | - | 16.9 ( 1.0) |
| | yes | SPDBN [4] | -1.6 ( 2.2) | -7.8 (0.0001) | 20.3 ( 1.6) |
| | no | SPDMBN (proposed) | -3.9 ( 4.4) | -10.7 (0.0001) | 11.3 ( 0.5) |
| no | yes | MBN [2] | -4.5 ( 3.8) | -10.1 (0.0001) | 6.6 ( 0.2) |
| | no | MBN [2] | -6.9 ( 4.8) | -13.4 (0.0001) | 4.4 ( 0.1) |

The performance gain comes at the cost of a 2.6x longer time to fit the parameters. The second largest effect can be attributed to DSBN; without DSBN the performance dropped by 3.9% (with $\mathcal{S}_D^+$) and 2.4% (w/o $\mathcal{S}_D^+$). The smallest, yet significant effect can be attributed to SPDMBN.

## 5.2 Mental workload estimation

Compared to the baseline methods, TSMNet obtained the highest average scores of 54.7% (7.3%) and 52.4% (8.8%) in inter-session and -subject TL (for details see appendix A.3.1). Interestingly, the inter-session TL score of TSMNet matches the score (54.3%) of the winning method in last year's competition [15]. To shed light on the sources utilized by TSMNet, we show patterns for a fitted model in Figure 3b. For the low mental workload class, the top contributing source's activity peaked in the theta band and originated in pre-frontal areas. The second source's activity originated in occipital cortex with non-focal spectral profile. Our results agree with the findings of previous research, as both areas and the theta band have been implicated in mind wandering and effort withdrawal [60].

## 6 Discussion

In this contribution, we proposed a machine learning framework around (domain-specific) momentum batch normalization on $\mathcal{S}_D^+$ to learn tangent space mapping (TSM) and feature extractors in an end-to-end fashion. In a theoretical analysis, we provided error bounds for the running estimate of the Fréchet mean as well as convergence guarantees under reasonable assumptions. We then applied the framework, to a multi-source multi-target unsupervised domain adaptation problem, namely, inter-session and -subject transfer learning for EEG data and obtained or attained state-of-the art performance with a simple, intrinsically interpretable model, denoted TSMNet, in a total of 6 diverse BCI datasets (138 human subjects, 573 sessions). In the case of mental imagery, we found that TSMNet significantly reduced (inter-subject TL) or even exceeded (inter-session TL) the performance gap to a SoA domain-specific method.

Although our framework could be readily extended to online UDA for unseen target domains, we limited this study to offline evaluations and leave actual BCI studies to future work. A limitation of our framework, and also any other method that involves eigen decomposition, is the computational complexity, which limits its application to high-dimensional SPD features (e.g., fMRI connectivity matrices with fine spatial granularity). Altogether, the presented results demonstrate the utility of our framework and in particular TSMNet as it not only achieves highly competitive results but is also intrinsically interpretable. While we do not foresee any immediate negative societal impacts, we provide direct contributions towards the scalability and acceptability of EEG-based healthcare [1, 5] and consumer [18, 60] technologies. We expect future works to evaluate the impact of the proposed methods in clinical applications of EEG like sleep staging [80, 81], seizure [82] or pathology detection [83, 84].

## Acknowledgments and Disclosure of Funding

This work was supported by the JSPS KAKENHI (Grants-in-Aid for Scientific Research) under Grant Numbers 20H04249, 20H04208, 21H03516 and 21K12055, and the Agency for Medical Research and Development (AMED) under Grant Number JP22dm0307009.

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
