# OpenReview forum: "SPD domain-specific batch normalization to crack interpretable unsupervised domain adaptation in EEG"
_NeurIPS.cc/2022/Conference — NeurIPS 2022 Accept_

### Official Review · Reviewer_P4Fn · 2022-07-07

**Rating:** 7
**Confidence:** 4
**Soundness:** 2 fair
**Presentation:** 3 good
**Contribution:** 3 good

**Summary:**

The authors proposes to learn domain invariant tangent space mapping models in an end to end fashion.
This mapping is suited to problems on data matrices that come from different domains and thus have possible different distributions.
These domain shifts occur in EEG when data are recorded on different people (structural and functional differences in brain networks) or several sessions (differences in the positioning of the electrodes).
The proposed differentiable pipeline is composed of 3 steps.
1) Data matrices, associated with different domains, are transformed to covariance matrices. The latter belong to the set of symmetric positive definite (SPD) matrices which is viewed as a Riemannian manifold with an affine invariant Riemannian metric.
2) The first and second order statistics of these matrices (grouped by domain) are normalized using a proposed Algorithm called "SPD domain-specific momentum batch normalization" (SPDDSMBN). Then the obtained matrices are transformed to vectors by lifting them to the tangent space at the identity using the logarithmic mapping.
3) Obtained vectors in the tangent space (linear space) are classified using a linear classifier.

The main contribution is the Algorithm SPDDSMBN. Covariance matrices of each domain are assumed to follow a Gaussian distribution, on the SPD manifold, parametrized by a mean covariance matrix (Fréchet mean) and a positive scalar (Fréchet variance).
SPDDSMBN estimates these parameters iteratively using batchwise estimates. The authors claim that the estimate of the Fréchet mean converges to its true value as the number of iterations tends to the infinite (Proposition 2). Then, the covariance matrices are transformed so that they have new learned first and second order moments. The interest is when the learned moments are enforced to be equal: all the covariance matrices from the different domains share the same statistics. This step is developed to improve the classification between different domains.

A second contribution is that the full pipeline is differentiable and thus is trained in a end-to-end manner. By choosing linear operators in the pipeline, the authors are able to derive a log-linear relationship between the targets and the variances of the sources the classical EEG generative model. Thus the discriminative sources are identifiable which makes it an interpretable method.

The proposed method is applied on 7 EEG datasets of mental imagery and mental workload estimation shows overall great performance compared to other classifiers.


**Questions:**

The disappointing part of this paper is about the convergence of the running mean to the Fréchet mean.

1) Can you clarify the confusion around $Var_{\theta_k}(\mathcal{T}_{\theta_k})$

which is sometimes referred as $Var_{\theta}(\mathcal{T}_{\theta_k})$ ?

Are they the same thing ? If not, $Var_{\theta}(\mathcal{T}_{\theta_k})$ should be defined.

2) Can you prove that if $f_\theta$ is $L$-smooth then (14) is respected ?

3) A comment should be added on why the induction in the proof of the Proposition 1 implies that the mapping $l \mapsto Var_{\theta_k}(G_l)$ is non-increasing.

4) Can you detail how (19) gives (29) ? The variance of $B_k$ should be compared with the variance of $\mathcal{T}_{\theta_k}$.

5) A comment on why the Proposition 1 is interesting/useful should be added. Is  it useful to state the Proposition 2 ?

6) The assumptions of the Proposition 2 should be stated explicitly (and not "If we additionally ...") and made clearer. If the Proposition 2 is new (versus the work of [50]), then a proof should be added. Otherwise, it can be transformed to a remark on how the Theorem of large numbers of [50] applies to the pipeline.

7) The introduction of many different notations makes the paper hard to read,

- e.g. the estimated batch mean is denoted $\hat{G}_\mathcal{B}, B_k$,

- the estimated Fréchet mean $G_k, G_\mathcal{T}(k)$,

- the Fréchet mean $G_{\theta_k}, \text{FM}(\mathcal{Z})$,

- the Riemannian distance $\delta, \delta_\text{AIRM}$ ... This should be simplified.

8) Minor remarks:
- In the section "Riemannian geometry on $\mathcal{S}^+_D$: the inner product $\langle ., . \rangle_Z$ is wrong .
- Equation (2): the logarithmic mapping should be written between two different matrices.
- After equation (3) $\mathcal{S}^+_d$ is written instead of $\mathcal{S}^+_D$.
- The parallel transport is used many times and thus should be presented in the section "Riemannian geometry on $\mathcal{S}^+_D$.
- Just before the equation (23): $G_{k-1}$ is written instead of $B_{k-1}$.

**Limitations:**

The authors mention the potential of the EEG applications to aid the society. Some references or one/two sentences could be added to justify.

**Strengths And Weaknesses:**

- Strengths:

Originality:

The proposed method is novel to me and leverages recent advances made on the SPD manifold such as the Gaussian distribution, the geometric law of large numbers, and parallel transport of SPD matrices. Recalibrating the second order moment is rarely done whereas it is easy to implement as shown in this paper.
Also, the proposed method leverages the recent advances on geometric deep learning (SPDnet) by training a full pipeline in an end-to-end manner. This should improves the performance compared to classical methods that perform sequential operations associated with different optimization problems.
The combination of these contributions is original.

Quality:

The numerical experiments are numerous and of great quality. The authors perform experiments on 7 EEG datasets of mental imagery and mental workload estimation and the proposed pipeline is compared to 10 other methods. They achieve state of the art performance on several of them and are on-par with best models on the others. Also, p-values of statistical tests are reported to ensure the significance of the reported results. An ablation study is performed and shows the importance of using all the proposed ingredients (momentum and domain-specific batchnorn). An additional good point is the Figure 3 which gives an interpretation of the learned model on two datasets.

Clarity:

The experimental part is clear: datasets are presented and the proposed method  used in practice is presented in Figure 4 and Table 3.


- Weaknesses:

Quality and clarity:

The theoretical part about the convergence of the running mean to the Fréchet mean is poorly written.
First of all, it seems that there is a confusion around
$Var_{\theta_k}(\mathcal{T}_{\theta_k})$

which is sometimes referred as $Var_{\theta}(\mathcal{T}_{\theta_k})$.

It seems that the latter is never defined.
Then, it is not very clear why the Proposition 1 (which is the only proven theoretical result) is mentioned. It is not interpreted and is never really used in the paper.
Furthermore, to prove this result the assumption (14) is made. The authors mention that it is met when $f_\theta$ is $L$-smooth. To me, this statement is not trivial and should be proven.
Also, the proof of the Proposition 1 is questionable. First of all, it is not clear why proving this induction proves that the mapping $l \mapsto Var_{\theta_k}(G_l)$ is non-increasing.
Then, it is not obvious how the equation (29) is obtained. The authors leverage the equation (19), but how does (19) (which relates variances of $\mathcal{T}_{\theta_k}$

between two successive iterates)
implies inequalities on the variances of $B_{k-1}$ and $B_k$. It is non-trivial since $B_k$ is an estimator (1-step Karcher flow) of the Fréchet mean.
The assumptions of the Proposition 2 are not very clear. Why do the previous assumptions such as the equation (14) are necessary ? Also, it seems to me that $\theta_k$ converges to a stationary point does not imply that $||\theta_k - \theta^\star|| \leq 1/k^\beta$. Furthermore, why does the convergence rate of $\theta$ must be in $1/k^\beta$ ? If after a certain amount of iterations, the $\theta_k$ stay in a small ball around $\theta^\star$, the covariance matrices $f_{\theta_k}(x)$ can be considered fixed and thus the Theorem of large numbers on the SPD manifold directly applies.


Finally, too many different notations are introduced in the paper which make it hard to read.

---

> ### Author Response · Authors · 2022-08-01
> **Response to official review of paper10594 by reviewer P4Fn [1/2]**
>
> Thank you very much for appreciating our proposed contribution, and in particular for the positive feedback about originality and quality of the numerical experiments.
> Please find below our responses to your comments and questions.
>
> >**WEAKNESSES**
>
> >The theoretical part about the convergence of the running mean to the Fréchet mean is poorly written.
>
> We are very sorry for the poor quality of the theoretical part. Thank you very much for your detailed comments and suggestions. Based on your feedback, we decided to restructure and extend the theoretical part in the revised contribution. Specifically, we use now consistent notation for the same concept, modified the assumption for proposition 1 to directly relate $Var_{\theta_k}(B_{k-1})$ with $Var_{\theta_k}(B_k)$, add interpretations of the results with regard to the batch normalization algorithms during early and late learning stages, and unified notation.
>
> >**Questions**
> >The disappointing part of this paper is about the convergence of the running mean to the Fréchet mean.
>
> Again, we are sorry for the poor quality of this part. We hope with our response to your questions and updates in the revised manuscript we can considerably improve its quality.
>
> >**(Q1)**: Can you clarify the confusion around $Var_{\theta_k} ( T_{\theta_{k}} )$ which is sometimes referred as $Var_{\theta}(T_{\theta_k})$? Are they the same thing ? If not, $Var_{\theta}(T_{\theta_k})$ should be defined.
>
> They actually refer to the same concept ($Var_{\theta_k} ( T_{\theta_{k}} ) = Var_{\theta}(T_{\theta_k}) $). We are very sorry for introducing the confusion due to inconsistent notation.
> Following your comment (4), we decided to base the assumption directly on the variance of the batch mean $Var_{\theta_k}(B_k)$. As a result, we do not use $Var_{\theta_k} ( T_{\theta_{k}} )$ in the revised manuscript anymore.
>
> >**(Q2)**: Can you prove that if f  is $L$-smooth then (14) is respected ?
>
> Since, we modified assumption (14), we would need to show that if $f_\theta$ is $L$-smooth then the variance of the previous batch mean$Var_{\theta_k}(B_{k-1})$ is bounded by the current one$Var_{\theta_k}(B_k)$. Although intuitively smoothness of $f_\theta$ with small steps seems sufficient for assumption (14), our attempts to prove that this is actually the case were unsuccessful, in the last days. We updated the text in the revised manuscript accordingly.
>
> >**(Q3)**: A comment should be added on why the induction in the proof of the Proposition 1 implies that the mapping $l \mapsto Var_{\theta_k}(G_l)$ is non-increasing.
>
> Thank you very much for this comment. This is indeed an important point. In the proof we show that $Var_{\theta_k}(G_k)$ is less or equal to $Var_{\theta_k}(B_k)$. However, from this proof it does not follow that $Var_{\theta_k}(G_k)$ is non-increasing over $k$. Example: due to learning, the feature extractor could increase the dataset’s Fréchet variance from one training step to another. As a result, the variance of the estimated batch mean $Var_\{theta_k}(B_k)$ will be larger than $Var_{\theta_{k-1}}(B_{k-1})$.
> As $Var_{\theta_k}(B_k)$  directly affects $Var_{\theta_k}(G_k)$ we cannot rule out that $Var_{\theta_k}(G_k) \le Var_{\theta_{k-1}}(G_{k-1})$.
>
> During rethinking about the intention and purpose of proposition 1 in the manuscript (see also your comment (5)), we identified that it is actually sufficient for our purposes to show that $Var_{\theta_k}(G_k)$ is less or equal to $Var_{\theta_k}(B_k)$ across training steps $k$ (details see response to your comment (5)).
>
> >**(Q4)**: Can you detail how (19) gives (29) ? The variance of $B_k$ should be compared with the variance of $T_k$.
>
> Thank you for identifying this issue. It helped us to refocus proposition 1. With our current approach (as detailed in our response to your comment 1), we think this issue is resolved.
>
> >**(Q5)**: A comment on why the Proposition 1 is interesting/useful should be added. Is it useful to state the Proposition 2?
>
> Proposition 1 is especially useful during initial learning (where the parameters of the model change quickly, and, thereby, also the Fréchet means. It provides bounds for the parameter updates so that SPD(M)BN can keep track of the changing Fréchet means. Proposition 1 specifically establishes a relationship between the parameter updates (learning $f_\theta$) and the momentum parameter (tracking the dataset’s Fréchet mean) so that the variance of the introduced noise though updating $G_k$ would be bounded by the variance of $B_k$. Furthermore, it explicitly shows how the introduced noise can be reduced, namely, via reducing the learning rate and momentum parameter.
> We added this information for the motivation and interpretation of proposition 1 in the revised manuscript.
>
> About your question, proposition 1 is not directly required to state proposition 2 (=remark 1 in the revised manuscript). Remark 1 is targeted towards a late stage of learning (i.e., Fréchet mean is changing only marginally).

---

> ### Author Response · Authors · 2022-08-01
> **Response to official review of paper10594 by reviewer P4Fn [2/2]**
>
> >**(Q6)**: The assumptions of the Proposition 2 should be stated explicitly (and not "If we additionally ...") and made clearer. If the Proposition 2 is new (versus the work of [50]), then a proof should be added. Otherwise, it can be transformed to a remark on how the Theorem of large numbers of [50] applies to the pipeline.
>
> As you suggested we transformed proposition 2 into remark 1. We also liked your reasoning, which assumes that the parameters stay within a ball around a fixed point (equation (17) in the revised manuscript), more than our previous approach, which required fast convergence. Combining the revised assumption (14) and  new assumption (17) imply that the Fréchet mean can be considered fixed. Then, the Theorem of large numbers on the SPD manifold is applicable.
>
> >**(Q7)**: The introduction of many different notations makes the paper hard to read.
>
> After some time has passed and with your and the other reviewers’ comments at hand, we certainly agree that the different notations are unnecessary and make the manuscript hard to read. We are sorry for not fixing this in the original manuscript. In the revised version, we stick to one notation.
>
> >e.g. the estimated batch mean is denoted $\hat{G}_B$, $B_k$
>
> We now only use $B_k$
>
> >the estimated Fréchet mean $G_k$, $G_\mathcal{T}(k)$
>
> We now only use $G_k$
>
> >the Fréchet mean $G_{\theta_k}$,$FM(Z)$
>
> We now only use $G_{\theta_k}$ when we consider a model with a feature extractor, or $G_\mathcal{A}$ when we refer to the Frechet mean of dataset $\mathcal{A}$.
>
> >the Riemannian distance $\delta$, $\delta_{AIRM}$... This should be simplified.
>
> Since we only use the affine invariant Riemannian metric (AIRM) throughout the manuscript, we now only use $\delta$.
>
> >Minor remarks:
>
> Thank you for identifying these mistakes and reporting them to us.
>
> >**Limitations**:
> >The authors mention the potential of the EEG applications to aid the society. Some references or one/two sentences could be added to justify.
>
> Thank you for this suggestion. We added a few references to works that highlight the potential of EEG in rehabilitation and healthcare in the first paragraph of section 5.
>
> Thank you very much for all your valuable comments. We hope that we could resolve your issues with our response in the revised manuscript, and if not yet, look forward to keep discussing with you to further improve the quality of our submission.
> For your information, the revised manuscript, incorporating the feedback of all reviewers can be accessed with this link: https://openreview.net/pdf?id=pp7onaiM4VB

---

> ### Comment · Reviewer_P4Fn · 2022-08-03
> **Remarks/questions about the rebuttal**
>
> Thank you for the different answers ant the revised version of the paper.
>
> After reading the revised version, I still have some remarks/questions:
>
> - Strengths:
> 1) The notations have been simplified and the paper is now much clearer.
> 2) The presentation of the different batch normalisations is easier to follow.
> 3) The Proposition 1 is also much clearer and shows the interest of considering running mean $G_k$.
>
> - Weaknesses:
> 1) Unfortunately, the statement that, if $f_\theta$ is smooth in the parameters and with small learning rates then the assumption (12) is respected, is transformed to a conjecture. Thus, it is hard to see the practical interest of this assumption which limits the scope of the first proposition.
> 2) The brace in (24) is wrong: it is $Var(U)$.
> 3) In order to get the inequality (31), you make the implicit assumption that the Karcher expectation of $G_k$ and $B_k$ is the Fréchet mean $G_{\theta_k}$. To me, this is not obvious and should be proved. Indeed, $G_k$ and $B_k$ are not necessarily unbiased estimators of $G_{\theta_k}$ in the Riemannian sense.
> 4) The sentence around the equation (15) is strangely written. In particular, (12) and (15) do not imply the approximation $Var(B_{k-1}) \approx Var(B_k)$.
> 5) In the remark 1: why do we need $\gamma(k) = 1/k^\alpha$ ? In [50], they use $\gamma(k) = 1/(k+1)$.

---

> > ### Author Response · Authors · 2022-08-07
> > **Response: Remarks/questions about the rebuttal [1/4]**
> >
> > Thank you very much for appreciating our changes regarding presentation of the different batch normalization algorithms, simplification of notations and proposition 1.
> >
> > About your newly raised questions/concerns, we would like to address each individually below.
> >
> > > 1.Unfortunately, the statement that, if fθ is smooth in the parameters and with small learning rates then the assumption (12) is respected, is transformed to a conjecture. Thus, it is hard to see the practical interest of this assumption which limits the scope of the first proposition.
> >
> > Thank you for your summary. We plan to address this issue in future work. For this paper, the more important theoretical result is the connection to the conventional tangent space mapping methods which use the dataset's Fréchet mean (i.e., remark 1 which establishes convergence to the Fréchet mean).
> >
> >
> > > 2. The brace in (24) is wrong: it is Var(U).
> >
> > Thank you very much for finding this typo.

---

> > ### Author Response · Authors · 2022-08-07
> > **Response: Remarks/questions about the rebuttal [2/4]**
> >
> >
> > >3. In order to get the inequality (31), you make the implicit assumption that the Karcher expectation of $\\mathbf{G}\_k$ and $\\mathbf{B}\_k$ is the Fréchet mean $\\mathbf{G}\_{\\theta\_k}$. To me, this is not obvious and should be proved. Indeed, $\\mathbf{G}\_k$ and $\\mathbf{B}\_k$ are not necessarily unbiased estimators of $\\mathbf{G}\_{\\theta\_k}$
> > in the Riemannian sense.
> >
> > Thank you for this comment.
> > To get from (30) to (31) we us (24) only once for $\\mathbf{B}\_k$, while we used the definition of $\\mathrm{Var}\_{\\theta\_k}(\\mathbf{G}\_{k-1})$.
> > Starting from (30), we have:
> >
> > $$
> >   \\mathrm{Var}\_{\\theta\_k}(\\mathbf{G}\_k) \\le (1-\\gamma) \\mathrm{Var}\_{\\theta\_k}(\\mathbf{G}\_{k-1}) + \\gamma \\mathrm{Var}\_{\\theta\_k}(\\mathbf{B}\_{k}) - \\gamma (1-\\gamma) \\mathbb{E}\_{\\mathbf{G}\_{k-1}} \\{  \\mathbb{E}\_{\\mathbf{B}\_{k}} \\{  \\delta^2(\\mathbf{G}\_{k-1}, \\mathbf{B}\_k) \\} \\}
> > $$
> >
> > Using (24) with $\\mathbf{U}=\\mathbf{B}\_{k}$ and $\\mathbf{V}=\\mathbf{G}\_{k-1}$ to simplify the inner expectation in the last term, we get
> >
> > $$
> > \\mathbb{E}\_{\\mathbf{B}\_{k}} \\{  \\delta^2(\\mathbf{G}\_{k-1}, \\mathbf{B}\_k) \\} \\ge \\mathrm{Var}\_{\\theta\_k}(\\mathbf{B}\_k) + \\delta^2(\\mathbf{G}\_{k-1}, \\mathbf{G}\_{\\theta\_k})
> > $$
> >
> > where we used the assumption that the Karcher expectation of $\\mathbf{B}\_k$ is the Fréchet mean $\\mathbf{G}\_{\\theta\_k}$.
> > Putting this result into the last term we have:
> >
> > $$
> > \\mathbb{E}\_{\\mathbf{G}\_{k-1}} \\{  \\mathbb{E}\_{\\mathbf{B}\_{k}} \\{  \\delta^2(\\mathbf{G}\_{k-1}, \\mathbf{B}\_k) \\} \\} \\ge \\mathrm{Var}\_{\\theta\_k}(\\mathbf{B}\_k) + \\mathbb{E}\_{\\mathbf{G}\_{k-1}} \\{ \\delta^2(\\mathbf{G}\_{k-1}, \\mathbf{G}\_{\\theta\_k}) \\}
> > $$
> >
> > Applying the definition of $\\mathrm{Var}\_{\\theta\_k}(\\mathbf{G}\_{k-1})$ (as defined in equation 28), we get (31).
> >
> > Consequently, it remains to show that the Karcher expectation of $\\mathbf{B}\_k$ is the Fréchet mean $\\mathbf{G}\_{\\theta\_k}$.
> >
> > Outline of our informal proof that the Karcher expectation of $\\mathbf{B}\_k$ is the Fréchet mean $\\mathbf{G}\_{\\theta\_k}$.
> > 1. We recall that the Karcher flow algorithm performs a step along the negative gradient of $\\mathrm{Var}\_{\\mathcal{B}\_k}(\\mathbf{Z})$ at the initial value $\\mathbf{Z} \\in \\mathcal{S}\_D^+ $
> > 2. We show that the gradient is a noisy, unbiased estimate of the gradient of $ \\mathrm{Var}\_{\\mathcal{T}\_{\\theta\_k}}(\\mathbf{Z}) $ (i.e., the objective that is minimized by dataset's Frechet mean $\\mathbf{G}\_{\\theta\_k}$).
> > 3. Since, we use different iid initial values (i.e., the batch's arithmetic mean), we effectively run multiple noisy, unbiased gradient descent steps for $ \\mathrm{Var}\_{\\mathcal{T}\_{\\theta\_k}}(\\mathbf{Z}) $ in parallel when we compute the Karcher expectation $\\mathbb{E} \\{ \\mathbf{B}\_k \\}$. Due to the geodesic convexity of $ \\mathrm{Var}\_{\\mathcal{T}\_{\\theta\_k}}(\\mathbf{Z}) $, it follows that with an infinte amount of data, the Karcher expectation of $\\mathbf{B}\_k$ is $\\mathbf{G}\_{\\theta\_k}$.
> >
> > The informal proof is provided in the next comment.

---

> > ### Author Response · Authors · 2022-08-07
> > **Response: Remarks/questions about the rebuttal [3/4]**
> >
> > Informal proof that the Karcher expectation of $\\mathbf{B}\_k$ is the Fréchet mean $\\mathbf{G}\_{\\theta_k}$.
> >
> > SPDMBN performs one step of the Karcher flow algorithm (algorithm 1, line1). This corresponds to a step along the negative gradient of the cost function $\\mathrm{Var}\_{\\mathcal{B}\_k}(\\mathbf{Z})$ (6), which is for batch $\\mathcal{B}\_k$:
> >
> > $$
> > \\mathrm{Var}\_{\\mathcal{B}\_k}(\\mathbf{Z})
> > = 1/|{\\mathcal{B}\_k}|   \\sum\_{\\mathbf{Z}\_j \\in \\mathcal{B}\_k} \\delta^2(\\mathbf{Z}\_j,\\mathbf{Z})
> > $$
> >
> > where $\\mathbf{Z} \\in \\mathcal{S}\_D^+$ is the initial estimate at which the Karcher flow algorithm is started.
> >
> > The Karcher flow algorithm peforms a Riemannian gradient descent step [see e.g. reference 1, equation (28)]:
> >
> > $$
> > \\mathbf{B}\_k = \\mathrm{Exp}\_{\\mathbf{Z}}(-\\nabla\_R \\mathrm{Var}\_{\\mathcal{B}\_k}(\\mathbf{Z}))
> > $$
> >
> > where $\\nabla\_R \\mathrm{Var}\_{\\mathcal{B}\_k}(\\mathbf{Z})$ is the Riemannian gradient:
> >
> > $$
> > \\nabla\_R \\mathrm{Var}\_{\\mathcal{B}\_k}(\\mathbf{Z})
> > = - 1/|\\mathcal{B}\_k|   \\sum\_{\\mathbf{Z}\_j \\in \\mathcal{B}\_k} \\mathrm{Log}\_{\\mathbf{Z}}(\\mathbf{Z}\_j)
> > $$
> >
> > For the objective $ \\mathrm{Var}\_{\\mathcal{T}\_{\\theta\_k}}(\\mathbf{Z}) $ to estimate the entire dataset's Frechet mean $\\mathbf{G}\_{\\theta\_k}$, we would have:
> >
> > $$
> > \\nabla\_R \\mathrm{Var}\_{\\mathcal{T}\_{\\theta\_k}}(\\mathbf{Z})
> > = - 1/|\\mathcal{T}\_{\\theta\_k}| \\sum\_{\\mathbf{Z}\_j \\in \\mathcal{T}\_{\\theta\_k}} \\mathrm{Log}\_{\\mathbf{Z}}(\\mathbf{Z}\_j)
> > $$
> >
> > $$
> > \\nabla\_R \\mathrm{Var}\_{\\mathcal{T}\_{\\theta\_k}}(\\mathbf{Z})
> > = \\frac{|\\mathcal{B}\_k|}{|\\mathcal{T}\_{\\theta\_k}|} \\nabla\_R \\mathrm{Var}\_{\\mathcal{B}\_k}(\\mathbf{Z}) + \\frac{|\\mathcal{T}\_{\\theta\_k} \\setminus \\mathcal{B}\_k|}{|\\mathcal{T}\_{\\theta\_k}|} \\nabla\_R \\mathrm{Var}\_{ \\mathcal{T}\_{\\theta\_k} \\setminus \\mathcal{B}\_k} (\\mathbf{Z})
> > $$
> >
> > Rearanging terms, we get:
> >
> > $$
> > \\nabla\_R \\mathrm{Var}\_{\\mathcal{B}\_k}(\\mathbf{Z})
> > = \\frac{|\\mathcal{T}\_{\\theta\_k}|}{|\\mathcal{B}\_k|} \\nabla\_R \\mathrm{Var}\_{\\mathcal{T}\_{\\theta\_k}}(\\mathbf{Z}) - \\frac{|\\mathcal{T}\_{\\theta\_k} \\setminus \\mathcal{B}\_k|}{|\\mathcal{B}\_k|} \\nabla\_R \\mathrm{Var}\_{ \\mathcal{T}\_{\\theta\_k} \\setminus \\mathcal{B}\_k} (\\mathbf{Z})
> > $$
> > where the last term can be considered as additive noise.
> > That is, when we compute $\\mathbf{B}\_k$, we essentially take a step along a noisy estimate of the gradient of $\\mathrm{Var}\_{\\mathcal{T}\_{\\theta\_k}}(\\mathbf{Z})$.
> >
> > Since the elements $\\mathbf{Z}\_j \\in \\mathcal{T}\_{\\theta\_k}$ are assigned randomly to batches $\\mathcal{B}\_k$ of size $M$ without replacement, we have a finite amount of combinations.
> > We denote the set of all possible batches as $ \\mathcal{C}\_{\\mathcal{B}\_k} $ with cardinality $ | \\mathcal{C}\_{\\mathcal{B}\_k} | = \\binom{|\\mathcal{T}\_{\\theta\_k}|}{M} $.
> > The expected value of $ \\nabla\_R \\mathrm{Var}\_{\\mathcal{B}\_k}(\\mathbf{Z}) $ is then:
> >
> > $$
> > \\mathbb{E}\_{\\mathcal{B}\_{k}} \\{ \\nabla\_R \\mathrm{Var}\_{\\mathcal{B}\_k}(\\mathbf{Z}) \\}
> > = \\sum\_{\\mathcal{B}\_{k} \\in \\mathcal{C}\_{\\mathcal{B}\_k}} \\frac{1}{| \\mathcal{C}\_{\\mathcal{B}\_k} |} \\nabla\_R \\mathrm{Var}\_{\\mathcal{B}\_k}(\\mathbf{Z})
> > $$
> >
> > Since each element $\\mathbf{Z}\_j \\in \\mathcal{T}\_{\\theta\_k}$ will be picked $\\frac{| \\mathcal{C}\_{\\mathcal{B}\_k} |}{| \\mathcal{T}\_{\\theta\_k} |}$ times, the sum on the right hand side reduces to $\\nabla\_R\\mathrm{Var}\_{\\mathcal{T}\_{\\theta\_k}}(\\mathbf{Z})$.
> > Hence, for any initial point $\\mathbf{Z} $ the batch gradients are noisy, unbiased estimates of the dataset's gradient.
> > Consequently, $\\mathbf{B}\_k$ varies around the estimate that would be attained if all the data were available for the first Karcher flow step.
> >
> > In the first Karcher flow step, we actually use the arithmetic mean of the batch elements as initial value $\\mathbf{Z}$ .
> > That is, when we compute the Karcher expectation $\\mathbb{E} \\{ \\mathbf{B}\_k \\}$, we have multiple iid initial points from which we obtain noisy, unbiased estimates of $\\nabla\_R\\mathrm{Var}\_{\\mathcal{T}\_{\\theta\_k}}$.
> > Since the cost $\\mathrm{Var}\_{\\mathcal{T}\_{\\theta\_k}}$ is geodesically convex, and we follow the negative gradient with noisy estimates in each $\\mathcal{B}\_{k} \\in \\mathcal{C}\_{\\mathcal{B}\_k}$, the expected value of $\\mathbf{B}\_{k}$ will converge to $\\mathcal{G}\_{\\theta\_k}$ as the aomount of data goes to infinity.
> > That is, for an infinite amount of data (e.g., batches of iid observations sampled from the data generating process), we have $ \\mathbb{E} \\{  \\mathbf{B}\_{k} \\} = \\mathcal{G}\_{\\theta\_k}$ for all $k$.
> >
> > This concludes the informal proof.

---

> > ### Author Response · Authors · 2022-08-07
> > **Response: Remarks/questions about the rebuttal [4/4]**
> >
> > >4. The sentence around the equation (15) is strangely written. In particular, (12) and (15) do not imply the approximation $Var\_{\\theta\_k}(\\mathbf{B}\_{k-1}) \\approx Var\_{\\theta\_k}(\\mathbf{B}\_k)$.
> > >5. In the remark 1: why do we need $\\gamma(k) = 1/k^\\alpha$
> > ? In [50], they use $ \\gamma(k) = 1/(k + 1) $.
> >
> > Thank you for both comments. We intended remark 1 to cover the late stage of learning where the feature extractors have approximately converged (i.e., the latent representations stay within a ball).
> > We realize that our first take in the revision was not ideal.
> > Focusing on your original suggestion, we will change lines 194 to 200 (the part that concerns Q4 and Q5) to the following:
> >
> > ---
> >
> > If we consider a late stage of learning, and in particular assume that after a certain number of iterations $K$ the parameters stay in a small ball with radius $\\rho$ around $\\theta^*$:
> >
> > $$
> >   || \\theta\_k - \\theta^* || \\le \\rho ~~~\\forall~k > K
> > $$
> >
> > and the feature extractor is $L$-smooth in the parameters:
> >
> > $$
> > \\delta(f\_\\theta(\\mathbf{x}), f\_{\\tilde{\\theta}}(\\mathbf{x})) \\le L || \\theta - \\tilde{\\theta} || ~ \\forall \\mathbf{x} \\in \\mathcal{T} ~ \\forall \\theta , \\tilde{\\theta}
> > $$
> >
> > then the distances are bounded $\\delta(f\_{\\theta\_k}(\\mathbf{x}), f\_{\\theta^*}(\\mathbf{x})) \\le \\rho L $.
> >
> > **Remark 1** (Convergence of $\\mathbf{G}\_k$ for SPDMBN) If $\\rho L$ is neglibile compared to the dataset's variance, then the Fréchet mean and variance can be considered fixed, and the theorem of large numbers on $\\mathcal{S}^+\_D$ [50] applies directly. That is, if the momentum parameter is decayed with schedule $\\gamma(k) = 1/(k-K)$ $\\forall k > K$ the running mean $\\mathbf{G}\_k$ converges to the Fréchet mean $\\mathbf{G}\_{\\theta^*}$ in probability as $k \\rightarrow \\infty $.
> >
> > ---
> >
> > We hope that we could resolve all your initial and newly raised concerns with our responses, and if not yet, look forward to keep discussing with you to further improve the quality of our submission.
> >
> > ---
> > **References**
> >
> > [1] S. Fiori, “Learning the Fréchet Mean over the Manifold of Symmetric Positive-Definite Matrices,” Cogn Comput, vol. 1, no. 4, pp. 279–291, Dec. 2009, doi: 10.1007/s12559-009-9026-7.

---

> ### Comment · Reviewer_P4Fn · 2022-08-07
> **Updating the score**
>
> I thank the authors for answering all my questions.
>
> The rebuttal has greatly improved the paper by making it much clearer.
>
> Consequently, I update the scores that I assigned:
> - presentation: 1 -> 3
> - rating: 4 -> 7
>
> Hence, I now vote to accept the paper.

---

### Official Review · Reviewer_A1dj · 2022-07-11

**Rating:** 7
**Confidence:** 5
**Soundness:** 3 good
**Presentation:** 3 good
**Contribution:** 3 good

**Summary:**

**Update after rebuttal**: I thank the authors for their responses and must say that I'm very happy with their rebuttal. Their submission has improved in this new version and I will, therefore, increase my score from borderline-accept (5) to accept (7).

---
This work presents a pipeline for classification with EEG recordings based on three steps. Firstly, the statistical features of the EEG time series are encoded via a symmetric positive definite (SPD) matrix. Then, they go through a neural network specially tailored for handling data points defined in the SPD manifold. Finally, the transformed SPD matrices are projected onto the tangent space of their manifold where a linear classifier with softmax activations is applied.

The authors were particularly interested in the case where the statistics of the training dataset is different from that of the testing set. This happens quite often in EEG, for instance when working with brain-computer interfaces trained on different subjects. The differences in these statistics is an important challenge in BCI research and the paper tackles it with their domain-specific version of the momentum batch normalization for SPD matrices (SPDDSMBN), which is a layer in their neural network. The authors also include two interesting features to classification pipeline: (1) the ability of learning from the data which SPD features should encode the time series, as opposed to the traditional approach of fixing them to be the spatial covariance matrix. (2) interpretability of the linear weights used to classify the data points, directly relating them to the spatial distribution of the EEG recordings on different channels.

**Questions:**

**(Q)** In the case where you have several source datasets, you simply pool all of their data points into a single big source dataset post transformations via SPDDSMBN? Does your procedure learn somehow to give more importance to data from one subject as compared to another one? In fact, your TSMNet is the same for all possible target subjects? Or you train one for each possible target?

**(Q)** Is there any theoretical result showing the limitations of SPDDSMBN to correct the dissimilarities between the statistics of the source and target dataset? In other words, do you have any mathematical model showing how a source and a target dataset may be different and whether your procedure is capable of correcting it?

**(Q)** What is the SOA domain-specific method used as reference for comparisons in Section 5.1?

**(Q)** In Equation 17 you present a layer depends on parameter $\phi$ and modulates the dispersion of the data points at its output. How
does your training procedure avoid making $v_\phi \to 0$ and decreasing the variance of its output as small as possible?

**(Q)** Saying "latent observations" in Line 172 is an oxymore. Do you observe these data points or not?

**(Q)** I am more accustomed to seeing the term "intra-domain" than "domain-specific". Do the authors have any specific reason for preferring this term?

**(Q)** Is Equation 7 really necessary for the rest of the paper?

**Limitations:**

The authors show the limitations of their method with applications to real data and show that while it does not always attain the same performance as a domain-specific SOA method, it does beat all other concurrent methods from the literature. However, I would have appreciated seeing a simulated example with controlled dissimilarities between the datasets and checking which class of transforms the method proposed by the authors is able to correct.

**Strengths And Weaknesses:**

As an expert on the domain of EEG classification with SPD matrices, I must say that I enjoyed seeing the many new ideas presented in this paper. There are several small gems that tackle different practical challenges of modern BCI systems and I congratulate the authors for getting interested in them. However, the sequence of the presentation of these ideas is quite confusing and it was not particularly easy to follow through the logic of the text, even for someone with some experience in the field.

For instance, it feels a bit odd to talk so much about batch normalization (and its variants) without saying why we need it for this application in the first place. Also, while I understand the structure of the architecture of TSMNet with SPDDSMBN, I still don't get why the theoretical results presented in Proposition 1 and Proposition 2 were necessary. Indeed, it feels much more like a math result that the authors wanted to include in the text than something that had a motivation for being part of the work (Please correct me if I'm mistaken).

In line 161, the authors say "we believe that the performance can be improved by extending MBN to the SPD manifold". This is a very strange and imprecise of presenting something that is supposed to be one of your main contributions. Also, I'm uncertain of what you mean by proposing a "theory-based" machine learning framework.

Also, the text can be a bit hard to read due to its (way too) many acronyms. Are the authors certain that all of them are really necessary? Things like MSMTUDA and SPDDSMBN can be a bit disturbing to read.

In all, the content of the paper is interesting and worth of being shared with the community, but I am not satisfied with the structure of the presentation of ideas and think that the authors should improve it before publication.

---

> ### Author Response · Authors · 2022-08-01
> **Response to official review of paper10594 by reviewer A1dj [1/4]**
>
> Thank you very much for your feedback. Please find our answers to your comments and questions below.
>
> >**Strengths And Weaknesses**:
>
> >As an expert on the domain of EEG classification with SPD matrices, I must say that I enjoyed seeing the many new ideas presented in this paper. There are several small gems that tackle different practical challenges of modern BCI systems and I congratulate the authors for getting interested in them. However, the sequence of the presentation of these ideas is quite confusing and it was not particularly easy to follow through the logic of the text, even for someone with some experience in the field.
>
> We are delighted that you appreciate the ideas presented in our submitted manuscript. After seeing your comments and reading the manuscript again, we agree that the way how we introduced the different methods - particularly section 3 and the last paragraph in the introduction - can be confusing. In the revised manuscript, we restructured both. First SPDMBN is introduced, then its theoretical analysis and its connection to TSM. At last, we introduce SPDDSMBN, which is used in section 4 to form TSMNet.
>
> >For instance, it feels a bit odd to talk so much about batch normalization (and its variants) without saying why we need it for this application in the first place.
>
> We extended the first paragraph in the introduction to motivate why we use batch normalization (track the domains’ Frechet means as they change during learning). We hope this resolves this issue.
>
> >Also, while I understand the structure of the architecture of TSMNet with SPDDSMBN, I still don't get why the theoretical results presented in Proposition 1 and Proposition 2 were necessary. Indeed, it feels much more like a math result that the authors wanted to include in the text than something that had a motivation for being part of the work (Please correct me if I'm mistaken).
>
> We think that this concern overlaps with the feedback of reviewer P4Fn. Thank you both. The theoretical part is required to provide bounds for the parameter updates so that SPD(M)BN can track the changing Frechet means (Proposition 1), and also establish a connection to TSM (=proposition 2 in the original submission; remark 1 in the revised manuscript). After incorporating the feedback in the revised manuscript, we think that the connection between the theoretical results and the rest is much more apparent.
>
> >In line 161, the authors say "we believe that the performance can be improved by extending MBN to the SPD manifold". This is a very strange and imprecise of presenting something that is supposed to be one of your main contributions.
>
> Thank you for pointing us to this strange formulation. We cut this part in the revised manuscript due to the restructuring of section 3.
>
> >Also, I'm uncertain of what you mean by proposing a "theory-based" machine learning framework.
>
> We used the term “theory-based” to refer to the theoretical results concerning the proposed SPDMBN layer in proposition 1 and proposition 2 (=remark 1 in the revised manuscript). As we also propose TSMNet to address the inter-session/-subject transfer learning problem in EEG without performing any theoretical analysis concerning the data alignment part, we decided to remove the term “theory-based” in the abstract.
>
> >Also, the text can be a bit hard to read due to its (way too) many acronyms. Are the authors certain that all of them are really necessary? Things like MSMTUDA and SPDDSMBN can be a bit disturbing to read.
>
> Thank you (and the other reviewers) for raising this concern. In the revised manuscript we reduced the number of acronyms and unified the notation for the same mathematical constructs. We additionally introduce a table (Table 1 in the revised manuscript) that provides the acronyms of relevant batch normalization algorithms together with their key differences. We hope that this makes the paper easier to read.
>
> >In all, the content of the paper is interesting and worth of being shared with the community, but I am not satisfied with the structure of the presentation of ideas and think that the authors should improve it before publication.
>
> Thank you for seeing merit in our contribution. We hope that the changes will improve the presentation of ideas.

---

> ### Author Response · Authors · 2022-08-01
> **Response to official review of paper10594 by reviewer A1dj [2/4]**
>
> >**Questions:**
>
> >**(Q1)** In the case where you have several source datasets, you simply pool all of their data points into a single big source dataset post transformations via SPDDSMBN?
>
> Assuming that you use the term ‘source datasets’ as synonym for ‘source domains’, the answer is yes.
>
> >Does your procedure learn somehow to give more importance to data from one subject as compared to another one?
>
> This is a very interesting point. We think the answer is yes. However, as detailed below, we think that this is not a specific property of TSMNet but due to minibatch-based learning in neural nets.
> We did not explicitly give more importance to any source domain. This was implemented via balanced sampling of observations from source domains to form minibatches and equal weights in the loss aggregation across observations in the minibatches.
> Knowing that there are large differences in the separability of the classes across subjects (see Figure 1c for an example), the minibatch based optimization process of neural nets could potentially downweight the contribution of less discriminative domains (=sessions of subjects) during model training. Specifically, the gradients of a minibatch with data from weakly discriminative domains will yield inconsistent gradients, and via aggregation likely reduce the magnitude of the parameter updates. Hence, for a fixed step size the model parameter updates will be smaller than for a minibatch containing data of highly discriminative domains. As this is the case for any minibatch based learning procedure any neural network architecture like EEGNet and ShConvNet would benefit from this effect.
>
> >In fact, your TSMNet is the same for all possible target subjects? Or you train one for each possible target?
>
> The TSMNet is the same for all possible target subjects. At test time, for each target domain (=session of a target subject) a new set of domain-specific statistics (Frechet mean and variance) are fitted to the target domain data inside the SPDDSMBN layer.
> Offline (i.e., the entire target domain’s data is revealed) this is done via projecting the data to the latent SPD domain with f_\theta and then using the Karcher flow algorithm to estimate the domain’s Frechet mean and variance. Online (i.e., in the case the target domain data is provided as a stream) the update rules of SPDMBN (algorithm 1) can be used to keep track of the stream’s Frechet mean and variance.

---

> ### Author Response · Authors · 2022-08-01
> **Response to official review of paper10594 by reviewer A1dj [3/4]**
>
> >**(Q2)** Is there any theoretical result showing the limitations of SPDDSMBN to correct the dissimilarities between the statistics of the source and target dataset? In other words, do you have any mathematical model showing how a source and a target dataset may be different and whether your procedure is capable of correcting it?
>
> Thank you for raising this interesting point. We focused our theoretical analysis on the convergence of the running mean of SPDMBN to the Frechet mean so that we could establish a connection to the classical step-wise-way how tangent space mapping (TSM) models are trained.
> Since we could establish this connection, the alignment part of TSMNet is very similar to the work of Zanini et al. [1] who proposed an UDA method called RCT, which transports the SPD observations (in their case spatial covariance matrices) from the Frechet mean to vary around the identity matrix. In a follow-up work [2] the same group proposed Riemannian Procrustes Analysis (RPA) which can align the Frechet mean and variance, and, if some labeled observations are available for the target domains,  rotate the data so that the distance between class-wise means are minimized.
> Essentially the alignment part of TSMNet can perform similar operations as the UDA part of RPA. That is, TSMNet can compensate for shifts in the distributions across domains and also overall differences in dispersion. However, if the domains contain contradictory information, our alignment approach - like any other UDA approach - is destined to fail. This can be the case if transformations u_i that preserve the Frechet mean and variance of each domain i but change the conditional distribution of the labels $p_i(y|Z)$ act on the data. As we do not have access to $p_i(y|Z)$ we cannot invert u_i. An example transformation would be the rotation operation in RPA [2].
> Let us assume a single source domain s and a target domain t for simplicity. Because of differences in u_s and u_t, after aligning the data with our approach, the conditional distributions can be contradictory $p_s(y|Z_a) \ne p_t(y|Z_a)$ for $Z_a$ drawn from the aligned data. Thus, the larger the disagreement between the conditional distributions, the lower will be the performance of the model trained with the source domain. For further details and explanatory examples on this topic we would like to refer to section 3.2 in [3].
>
> [1] P. Zanini, M. Congedo, C. Jutten, S. Said, and Y. Berthoumieu, “Transfer Learning: A Riemannian Geometry Framework With Applications to Brain–Computer Interfaces,” IEEE Trans. Biomed. Eng., vol. 65, no. 5, pp. 1107–1116, May 2018, doi: 10.1109/TBME.2017.2742541.
>
> [2] P. L. C. Rodrigues, C. Jutten, and M. Congedo, “Riemannian Procrustes Analysis: Transfer Learning for Brain–Computer Interfaces,” IEEE Trans. Biomed. Eng., vol. 66, no. 8, pp. 2390–2401, Aug. 2019, doi: 10.1109/TBME.2018.2889705.
>
> [3] O. Yair, F. Dietrich, R. Talmon, and I. G. Kevrekidis, “Domain Adaptation with Optimal Transport on the Manifold of SPD matrices,” arXiv:1906.00616 [cs, stat], Jul. 2020, Available: http://arxiv.org/abs/1906.00616
>
> >**(Q3)** What is the SOA domain-specific method used as reference for comparisons in Section 5.1?
>
> Due to the success of TSM models [4,5], we considered a spectrally resolved model which consisted of a filter-bank to separate activity of canonical frequency bands. For each frequency band, PCA was used to reduce the spatial dimensionality and TSM to project the SPD features to the Euclidean vector space. Finally, all features were pooled and submitted to a penalized logistic regression classifier. For further details, see [6]. The method is abbreviated as FB+TSM+LR in the manuscript and is also used as a baseline method in the transfer learning settings (Figure 2a).
> In the original manuscript, we briefly defined the domain-specific baseline model in the caption of Figure 2. As this information can be easily missed, we also added a sentence in section 5 and more details in the supplementary materials (appendix A.2.4 in the revised manuscript).
>
> [4] V. Jayaram and A. Barachant, “MOABB: trustworthy algorithm benchmarking for BCIs,” J. Neural Eng., vol. 15, no. 6, p. 066011, Dec. 2018, doi: 10.1088/1741-2552/aadea0.
>
> [5] D. Sabbagh, P. Ablin, G. Varoquaux, A. Gramfort, and D. A. Engemann, “Manifold-regression to predict from MEG/EEG brain signals without source modeling,” in Advances in Neural Information Processing Systems, 2019, pp. 7323–7334.
>
> [6] R. J. Kobler, J.-I. Hirayama, L. Hehenberger, C. Lopes-Dias, G. Müller-Putz, and M. Kawanabe, “On the interpretation of linear Riemannian tangent space model parameters in M/EEG,” 2021. doi: 10.1109/EMBC46164.2021.9630144.

---

> ### Author Response · Authors · 2022-08-01
> **Response to official review of paper10594 by reviewer A1dj [4/4]**
>
> >**(Q4)** In Equation 17 you present a layer depends on parameter $\phi$ and modulates the dispersion of the data points at its output. How does your training procedure avoid making $\nu_\phi \rightarrow 0$ and decreasing the variance of its output as small as possible?
>
> We did not implement any explicit measure to prevent $\nu_\phi$ from attaining the value 0 in TSMNet. So, in principle the model could learn to reduce$ \nu_\phi$ to 0. However, this will depend on the loss that is optimized. In our case, the standard cross-entropy loss would discourage $\nu_\phi$ from attaining the value 0  because if the Fréchet variance is reduced to 0, it will not be possible to discriminate the data, which, in turn, would result in a large loss.
>
> >**(Q5)** Saying "latent observations" in Line 172 is an oxymore. Do you observe these data points or not?
>
> Thank you for pointing us to this oxymoron. We replaced “observation” with “representation” in the revised manuscript.
>
> >**(Q6)** I am more accustomed to seeing the term "intra-domain" than "domain-specific". Do the authors have any specific reason for preferring this term?
>
> We decided to use “domain-specific” rather than “intra-domain” to be consistent with domain-specific BN [7] which is a key ingredient in our approach.
>
> [7] W.-G. Chang, T. You, S. Seo, S. Kwak, and B. Han, “Domain-Specific Batch Normalization for Unsupervised Domain Adaptation,” Jun. 2019.
>
> >**(Q7)** Is Equation 7 really necessary for the rest of the paper?
>
> No. As part of the restructuring that you suggested, we removed it in the revised manuscript.
>
> >**Limitations**:
> >The authors show the limitations of their method with applications to real data and show that while it does not always attain the same performance as a domain-specific SOA method, it does beat all other concurrent methods from the literature. However, I would have appreciated seeing a simulated example with controlled dissimilarities between the datasets and checking which class of transforms the method proposed by the authors is able to correct.
>
> In this paper we set the focus on batch normalization algorithms on the SPD manifold with first encouraging results (real EEG datasets) on the EEG transfer learning problem. We feel that the content of the paper is already quite dense at this stage, which led us to decide to leave in detail analysis of the UDA part (including simulations of the methods limitations) subject to future work.
>
> We hope that our responses and changes in the revised manuscript were sufficient to resolve your concerns; and if not yet, look forward to keep discussing with you to further improve the quality of our submission.
> For your information, the revised manuscript, incorporating the feedback of all reviewers can be accessed with this link: https://openreview.net/pdf?id=pp7onaiM4VB

---

> ### Comment · Reviewer_A1dj · 2022-08-07
> **Updating score**
>
> I thank the authors for their responses and must say that I'm very happy with their rebuttal. Their submission has improved in this new version and I will, therefore, increase my score from borderline-accept (5) to accept (7).

---

### Official Review · Reviewer_D8q8 · 2022-07-12

**Rating:** 8
**Confidence:** 3
**Soundness:** 4 excellent
**Presentation:** 4 excellent
**Contribution:** 4 excellent

**Summary:**

The authors present a new batch normalization module designed to improve performance of neural networks in scenarios where domain adaptation is critical, such as classification of inter-session and inter-subject EEG data. This is achieved by combining the ideas of momemtum BN, domain-specific BN, while respecting the manifold of SPD matrices.

On top of theoretical results supporting their approach, the authors provide compelling empirical evidence that the proposed SPDDSMBN module, when inserted into an SPD-aware deep neural network architecture, helps improve cross-domain performance compared to existing state-of-the-art approaches, as shown on 6 EEG datasets encompassing BCI and mental workload estimation classification tasks. An ablation study further confirms the importance of the proposed mechanisms for achieving these results.

**Questions:**

A single point that would make this manuscript even more convincing for me, though not necessary, would be if authors had provided results on a wider range of classification tasks, e.g. sleep staging, seizure detection, emotion classification and/or pathology detection. These tasks might present different inter-session and inter-subject variability characteristics. I believe including results on the mental workload estimation task was a first step in that direction. Whether results can extend to these types of tasks and even to other types of multivariate time series would be very interesting to see.

**Limitations:**

The authors have adequately addressed the limitations and negative impact of their work.

**Strengths And Weaknesses:**

The proposed SPDDSMBN method is original. While a logical extension of existing batch normalization ideas, it combines multiple important ideas that together provide significant performance improvements over existing state-of-the-art methods. Moreover, the interpretability of the proposed approach makes it particularly interesting for neuroscience-minded audiences who might want to explore correlates of the different classes in an EEG dataset.

The paper is of high quality and includes compelling theoretical proofs, experiments and results on multiple datasets. The manuscript is clearly written and well organized, as is the provided code.

Overall, I believe the manuscript has particular relevance to the NeurIPS community, even beyond the EEG and neuroimaging communities.

---

> ### Author Response · Authors · 2022-08-01
> **Response to official review of paper10594 by reviewer D8q8**
>
> Thank you very much for seeing merit in our submission and the very positive feedback.
>
> >A single point that would make this manuscript even more convincing for me, though not necessary, would be if authors had provided results on a wider range of classification tasks, e.g. sleep staging, seizure detection, emotion classification and/or pathology detection. These tasks might present different inter-session and inter-subject variability characteristics. I believe including results on the mental workload estimation task was a first step in that direction. Whether results can extend to these types of tasks and even to other types of multivariate time series would be very interesting to see.
>
> We also consider this work as the first step in evaluating the potential of our proposed framework to aid quantitative EEG analysis. Extending the proposed methods to other EEG application scenarios, like the ones that you mentioned, requires different assumptions (e.g., unbalanced class-priors in sleep spaying or seizure detection) that need to be addressed in future work.
>
> For your information, the revised manuscript, incorporating the feedback of all reviewers can be accessed with this link: https://openreview.net/pdf?id=pp7onaiM4VB

---

### Official Review · Reviewer_LpZK · 2022-07-15

**Rating:** 7
**Confidence:** 3
**Soundness:** 3 good
**Presentation:** 4 excellent
**Contribution:** 3 good

**Summary:**

The paper proposes a batch normalization framework that is suitable for problems of unsupervised domain adaptation (UDA) between symmetric positive definite (SPD) manifolds. Their framework- the SPD domain-specific momentum batch normalization or SPDDSMBN is designed to transform domain-specific SPD input matrices into domain-invariant SPD outputs by automatically inferring relevant batch statistics. This can be inserted a parameterized layer transformation within a geometric deep learning model in multi-(domain) source and target settings. The authors also provide a theoretical analysis on the correctness of recovered batch statistics. The authors perform experiments with 6 different EEG brain-computer interface (BCI) datasets and demonstrate that their framework provides improved performance within two transfer learning setups (inter-session and inter-subject) against baseline comparisons and ablation studies


**Questions:**

1. A suggestion would be to include more details on generating the interpretability maps from the reference [17] in the supplementary for completeness. Adding a few more sentences of explanation to the plots in Fig. 3 would also help improve the clarity here.
2. There are many acronyms used in the paper for each discussed variation of the framework. A suggestion would be to include a list of these for easy lookup within the supplementary document so a potential reader can avoid back and forth within the main text.



**Limitations:**

The main limitations of the framework, scalability to larger sized input data (eg. fMRI) is discussed briefly in Section 6. I agree that this is a fair criticism of all of the methods. Perhaps a relevant future direction would be to utilize techniques that can improve the scalability of the bottleneck eigenvalue decomposition.

**Strengths And Weaknesses:**

STRENGTHS:

1. The paper is very well written and easy to follow.  The relevant technical details are presented concisely and cover the relevant preliminaries making the paper fairly self contained. The main contributions are clearly articulated with dedicated theoretical analysis and thorough experimental evaluations provided to support the claims.

2. The main contributions of the paper, extending momentum based batch-normalization to geometric deep learning on SPD manifolds as well as the application to domain adaptation in SPD manifolds is interesting and relevant to applications beyond the EEG BCI domain. Although not a major contribution of the work, the discussion on interpretability in the context of the potential healthcare application is a plus.

WEAKNESSES:

There are very few details provided for generating the interpretability maps in Fig. 3 making this section a bit hard to parse.

(a) As per my understanding, the contribution of each extracted source to the target class (spatial mixing coefficient) is being plotted on the head model. I am a bit unclear on how to actually interpret the results being displayed, for example, are the darker regions more discriminative/important (the figure is also missing a colourbar)? This needs a bit of clarification
(b) I am unclear on what the axes of the plot on the top left of 3 a and 3 b are plotting. Specifically, what is the y axis "contribution" measuring?

---

> ### Author Response · Authors · 2022-08-01
> **Response to official review of paper10594 by reviewer LpZK**
>
> Thank you very much for your feedback. Please find our answers to your comments and questions below.
>
> >There are very few details provided for generating the interpretability maps in Fig. 3 making this section a bit hard to parse.
>
> Yes, that is true. Thank you for raising this issue. We provide more details in the revised manuscript and hope that this resolves your concern.
>
> >(a) As per my understanding, the contribution of each extracted source to the target class (spatial mixing coefficient) is being plotted on the head model.
>
> Yes, the topographic plots on the head model summarize the spatial pattern of the sources that are coupled to the target (equation 21 in the revised manuscript). However, the spatial pattern corresponds to the columns of the mixing matrix A, defined in the generative model (equation 20 in the revised manuscript). That is, each column of A tells us how the activity of a single source is projected to all EEG channels (=forward model).
> To identify which sources contributed most to the classification task, we ordered the sources according to their relative contribution to the target. As this part can be confusing, we provide a definition of the term “contribution” in the revised manuscript (equation 22) and an updated figure caption.
>
> >I am a bit unclear on how to actually interpret the results being displayed, for example, are the darker regions more discriminative/important (the figure is also missing a colourbar)?
>
> Yes, the darker areas (darker red or blue) indicate more discriminative regions because the sources whose activity is coupled to the target, have a stronger contribution to these regions (more precisely to the EEG channels that cover the regions). We added a colorbar in the revised manuscript.
>
> >This needs a bit of clarification (b) I am unclear on what the axes of the plot on the top left of 3 a and 3 b are plotting. Specifically, what is the y axis "contribution" measuring?
>
> In response to your first question, we expanded section 4 to briefly review the approach of 17 in the revised manuscript. We now provide a definition of the term “contribution” and refer to it in the caption of Figure 3.
> Questions:
>
> >A suggestion would be to include more details on generating the interpretability maps from the reference [17] in the supplementary for completeness. Adding a few more sentences of explanation to the plots in Fig. 3 would also help improve the clarity here.
>
> Thank you for the suggestion. We followed your suggestion and expanded the methods concerning the interpretability results. Since we think that this part is very important for EEG analysis, we decided to include this additional information in the main manuscript in section 4.
>
> >There are many acronyms used in the paper for each discussed variation of the framework. A suggestion would be to include a list of these for easy lookup within the supplementary document so a potential reader can avoid back and forth within the main text.
>
> Thank you (and the other reviewers) for raising this concern. In the revised manuscript we reduced the number of acronyms and unified the notation for the same mathematical constructs. We additionally introduce a table (Table 1 in the revised manuscript) that provides the acronyms of relevant batch normalization algorithms together with their key differences. We hope that this makes the paper easier to read. For your information, the revised manuscript can be accessed via this link: https://openreview.net/pdf?id=pp7onaiM4VB

---

> > ### Comment · Reviewer_LpZK · 2022-08-06
> > **Response to Authors' reply**
> >
> > I thank the authors for their reply and efforts in addressing my concerns. Their changes to the manuscript have improved the clarity and quality of presentation. I continue to vote to accept the paper.

---

### Author Response · Authors · 2022-08-01
**Revision uploaded**

Dear anonymous reviewers.

Thank you very much for your effort to review our submitted manuscript. We appreciate your feedback and tried to address all the issues and questions that you raised in our reply to your reviews and the updated manuscript.

The revised manuscript can be accessed with this link:
https://openreview.net/pdf?id=pp7onaiM4VB

---

### Meta-Review · Area_Chair_ppxQ · 2022-08-22

**Recommendation:** Accept
**Confidence:** Certain

**Metareview:**

The reviewers have now unanimously acknowledged the quality of the contribution both on the theory and experimental sides. This paper can be endorsed for publication at NeurIPS 2022.

**Award:**

No

---

### Decision · Program_Chairs · 2022-09-14

Accept